# Microglia are an essential component of the neuroprotective scar that forms after spinal cord injury

Victor Bellver-Landete[1], Floriane Bretheau[1], Benoit Mailhot[1], Nicolas Vallières[1], Martine Lessard[1], Marie-Eve Janelle[2], Nathalie Vernoux[1], Marie-Ève Tremblay [1], Tobias Fuehrmann [3], Molly S. Shoichet[3] & Steve Lacroix [1]

The role of microglia in spinal cord injury (SCI) remains poorly understood and is often confused with the response of macrophages. Here, we use specific transgenic mouse lines and depleting agents to understand the response of microglia after SCI. We find that microglia are highly dynamic and proliferate extensively during the first two weeks, accumulating around the lesion. There, activated microglia position themselves at the interface between infiltrating leukocytes and astrocytes, which proliferate and form a scar in response to microglia-derived factors, such as IGF-1. Depletion of microglia after SCI causes disruption of glial scar formation, enhances parenchymal immune infiltrates, reduces neuronal and oligodendrocyte survival, and impairs locomotor recovery. Conversely, increased microglial proliferation, induced by local M-CSF delivery, reduces lesion size and enhances functional recovery. Altogether, our results identify microglia as a key cellular component of the scar that develops after SCI to protect neural tissue.

[1] Axe neurosciences du Centre de recherche du Centre hospitalier universitaire (CHU) de Québec–Université Laval et Département de médecine moléculaire de l'Université Laval, Québec, QC G1V 4G2, Canada. [2] Département de biologie-biotechnologie du Cégep de Lévis-Lauzon, Lévis, QC, Canada G6V 6Z9. [3] Department of Chemical Engineering & Applied Chemistry, University of Toronto, Toronto, ON M5S 3E1, Canada. Correspondence and requests for materials should be addressed to S.L. (email: Steve.Lacroix@crchul.ulaval.ca)

Microglia is derived from primitive yolk sac progenitors that arise during embryogenesis[1–3]. They are maintained after birth and into adulthood by self-renewal[4,5], independently from bone marrow-derived hematopoietic stem cells (HSCs) and their differentiated progeny (e.g. monocyte-derived macrophages, MDMs)[6,7]. After a CNS injury, blood-derived monocytes are massively recruited in the tissue where they differentiate into macrophages and adopt many of the markers and behaviors of microglia. These similarities have complicated the development of efficient prediction tools to discriminate between them. As a consequence, they are still referred to as microglia/macrophages in the neuroscience literature, and accordingly, their individual roles remain to be clarified.

Recent advances in genetic fate mapping and conditional gene targeting have allowed the study of the specific biology of microglia in various experimental contexts, including spinal cord injury (SCI)[8]. This, together with the newly developed strategies to specifically eliminate microglia[9], has moved forward knowledge about these cells substantially. For example, the application of some of these advances to a mouse model of stroke has led to the discovery that microglia can protect neurons through the regulation of calcium levels[10]. In contrast, the elimination of microglia in mouse models of Alzheimer's disease and Tau pathology reduced disease progression[11,12]. Thus, depending on the context, microglia may exert diverging roles. Whether these cells are beneficial or deleterious after SCI remains unexplored.

Here, we took advantage of Cx3cr1creER mice[13], a mouse line that allows with an adequate regimen of tamoxifen to label microglia while excluding nearly all MDMs. Our results show that microglia are highly dynamic and proliferate extensively during the first week post-SCI. Notably, we reveal that microglia form a dense cellular interface at the border of the lesion between reactive astrocytes and infiltrating MDMs, which we hereafter refer to as the "microglial scar". Microglia depletion experiments using PLX5622, a CSF1R inhibitor that crosses the blood–spinal cord barrier (BSCB), demonstrated that the absence of microglia in the context of SCI disrupts the organization of the astrocytic scar, reduces the number of neurons and oligodendrocytes at the site of injury, and impairs functional recovery. The timing of the beneficial effects of microglia was estimated to be during the first week post-SCI. Accordingly, CNS delivery of M-CSF during that critical period boosted microglial proliferation and enhanced locomotor recovery. In light of these data, we conclude that microglia are an important component of the protective scar that forms after SCI.

## Results

**Microglia rapidly accumulate around the site of SCI**. To distinguish microglia from MDMs, we took advantage of Cx3cr1creER::R26-TdT mice, and the slow turnover of microglia[4,5]. Mice received tamoxifen treatment one month before SCI to activate the inducible Cre for recombination of TdT floxed (Supplementary Figure 1a). As expected from our previous work[14], nearly all (99.6 ± 0.2%) CD11b+ cells in the spinal cord parenchyma expressed TdT (Supplementary Figure 1b, c). In contrast, only a few CD11b+ cells in the blood, spleen and bone marrow were TdT+, with average colocalization percentages of 3.8 ± 1.7%, 6.7 ± 1.6%, and 2.4 ± 0.2%, respectively (Supplementary Figure 1d–f). Thus, inducible Cx3cr1creER::R26-TdT mice are a good tool to study microglia in SCI.

To understand the dynamics of the microglial response after SCI, we first quantified the total number of TdT+ microglia both in normal conditions and at 1, 4, 7, 14 and 35 days post-injury (dpi) (Fig. 1a–g and Supplementary Figure 2). In the uninjured thoracic spinal cord of Cx3cr1creER::R26-TdT mice, we counted an average of

85.9 ± 4.6 microglia per mm². Following a moderate contusive SCI, only 28.8 ± 1.9 microglia per mm² were left at the lesion epicenter at 1 dpi, which corresponds to a 67% reduction in cell numbers. Hardly any TdT+ microglia were observed in the lesion core at this early time point, suggesting that they underwent rapid cell death. Despite the fact that the impactor tip measures 1.25 mm of diameter, microglia were lost across several spinal cord segments rostrocaudally. This microglial cell loss ranged from ~20% to 65% at rostrocaudal distances up to 6 mm from the lesion epicenter (Fig. 1g, h), and was mediated in part through apoptosis (Fig. 1i–k). At that time, residual microglia still expressed the purinergic receptor P2ry12 (Supplementary Figure 3), a receptor implicated in microglia recruitment during the early acute phase of CNS injury[15]. Accordingly, we noticed a retraction of microglial processes as early as day 1. Expression levels of the lysosome-associated glycoprotein CD68, a marker of phagocytosis, remained low in TdT+ microglia at 1 dpi (Supplementary Figure 4). However, the situation changed at day 4, as we counted 119.1 ± 15.0 microglia per mm² at the lesion epicenter, which represents a four-fold increase in the number of TdT+ microglia compared to day 1 (Fig. 1g). Microglia around the lesion epicenter exhibited a round morphology, downregulation of P2ry12 and a strong upregulation of CD68 (Supplementary Figures 3 and 4), which points to a potential increase in their phagocytic activity starting around 4 dpi. The number of microglia continued to increase at the lesion epicenter over time, reaching up to 1204.61 ± 137.8 cells/mm² at 14 dpi. Nearly all microglia observed in these areas were TdT+ CD68hi P2Y12neg (Supplementary Figures 3 and 4). A similar trend was seen in the surrounding tissue (400–800 μm) of the lesion epicenter up to 35 dpi, after which the total number of microglia (i.e. 673.91 ± 62.4 cells/mm² at the lesion epicenter) started to decrease compared to day 14 (Fig. 1g). Interestingly, TdT+ microglia started to gradually increase their expression of P2ry12 and decrease their expression of CD68 from day 14 up to day 35 (Supplementary Figures 3 and 4), suggesting a partial return to homeostasis. In sum, our data indicate that microglia are rapidly recruited around the site of SCI, where they accumulate extensively during the subacute phase and adopt an activated state that is eventually partially resolved during the intermediate/chronic phases.

**Microglia proliferate extensively during the subacute phase of SCI**. Under normal circumstances, the adult microglial population remains stable in the brain throughout life by coupled cell death and cell proliferation[4], but little is known about its dynamic in the spinal cord and how it reacts following SCI. Here, we report that 0.6 ± 0.1 microglia/mm² (0.58% of total) are proliferating in the normal thoracic spinal cord of Cx3cr1creER::R26-TdT mice, as revealed by the co-expression of TdT and the cell proliferation marker Ki67. One day after SCI, no significant changes were observed in terms of microglial cell proliferation compared to the uninjured spinal cord (Fig. 1l, m). Strikingly, about 50% of microglia at the lesion epicenter expressed Ki67 at 4 dpi. As shown in Supplementary Movie 1, proliferating TdT+ Ki67+ microglia were round-shaped. The peak proliferation of microglia, in terms of absolute numbers, was observed at day 7 (Fig. 1l, n–p). At 14 and 35 dpi, only a few (2–6%) microglia were still expressing Ki67, suggesting that microglial proliferation was mostly inhibited by then. Together, these results show that microglia are highly dynamic after SCI, not only through their recruitment and activation, but also through their ability to rapidly proliferate and surround the site of SCI during the subacute phase.

**Depletion of microglia reduces locomotor recovery after SCI**. Yolk sac progenitors are at the origin of specialized tissue-resident

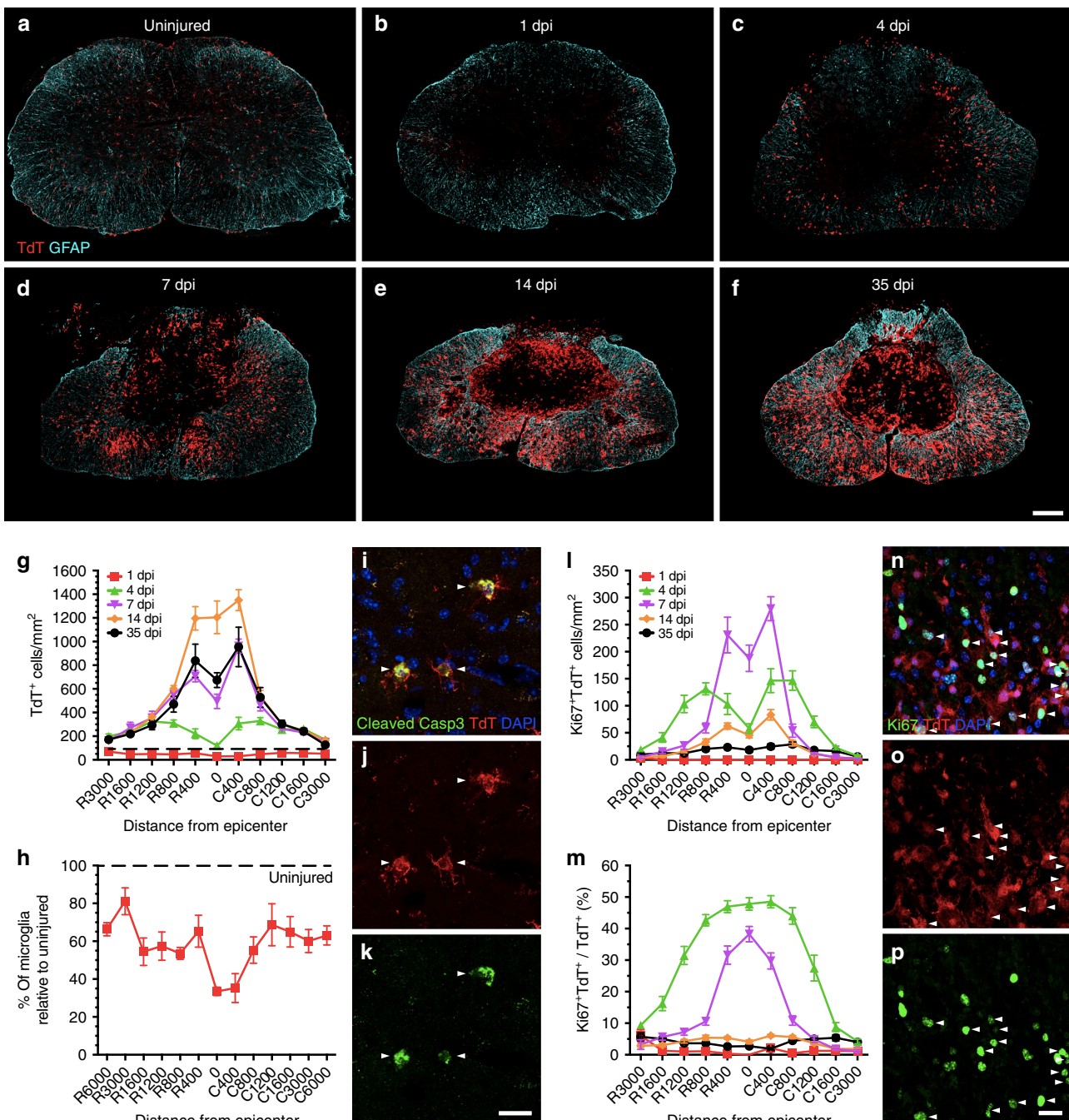

**Fig. 1** Microglia proliferate extensively and accumulate at the lesion border after SCI. **a**–**f** Confocal immunofluorescence microscopy of representative spinal cord sections showing the spatio-temporal distribution of microglia (TdT, red) and astrocytes (GFAP, cyan) in an uninjured *Cx3cr1*[creER]::*R26-TdT* transgenic mouse (**a**), as well as at the lesion epicenter at 1 (**b**), 4 (**c**), 7 (**d**), 14 (**e**), and 35 (**f**) days post-injury (dpi). **g** Quantification of the total number of TdT$^+$ microglia per mm$^2$ of tissue in spinal cord sections taken both rostral (R) and caudal (C) to the lesion epicenter at 1 (red line), 4 (green), 7 (violet), 14 (orange), and 35 (black) dpi ($n = 6$–8 mice per group/time point). Data from uninjured mice are shown with the dotted black line. **h** Percentage of surviving TdT$^+$ microglia at day 1 post-SCI relative to the uninjured group. **i**–**k** Confocal immunofluorescence images showing expression of the apoptotic marker cleaved caspase-3 (Casp3, green) in TdT$^+$ microglia (red) at 1 dpi. Nuclear staining (DAPI) is shown in blue, while white arrowheads indicate co-localization of cleaved Casp3, TdT, and DAPI. **l** Quantification of the number of actively proliferating microglia (TdT$^+$ Ki67$^+$ cells) at 1, 4, 7, 14, and 35 dpi ($n = 7$–8 mice per group). **m** Percentage of TdT$^+$ microglia undergoing proliferation after SCI ($n = 7$–8 per group). **n**–**p** Confocal immunofluorescence microscopy showing that TdT$^+$ microglia (red) are actively proliferating at 7 dpi, as demonstrated by their expression of the proliferation marker Ki67 (green). DAPI is shown in blue, while white arrowheads indicate co-localization of TdT, Ki67, and DAPI staining. Data are expressed as mean ± SEM. Scale bars: (**a**–**f**, in **f**) 200 μm; (**i**–**k**, in **k**) 20 μm; (**n**–**p**, in **p**) 20 μm

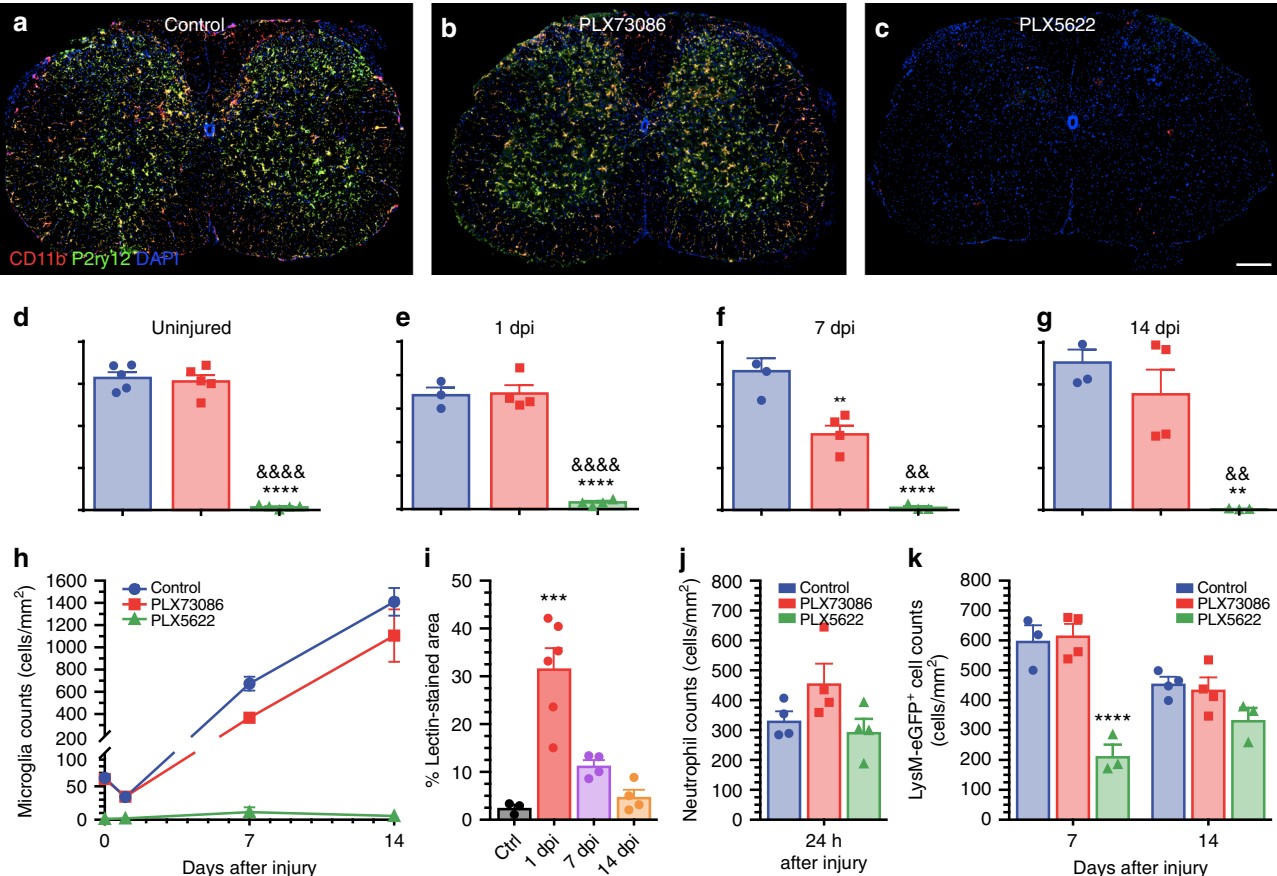

**Fig. 2** The CSF1R inhibitor PLX5622, but not PLX73086, crosses the blood–spinal cord barrier to deplete virtually all microglia. **a–c** Representative confocal images of CD11b and P2ry12 immunostainings showing the almost complete elimination of microglia in the spinal cord of naïve (uninjured) C57BL/6 mice after treatment with the CSF1R inhibitor PLX5622 compared to those fed PLX73086 or the control diet. Mice were killed after 21 days of treatment. **d–h** Quantification of microglia in the spinal cord of uninjured C57BL/6 mice treated with PLX5622, PLX73086 or the control diet (**d**), as well as at the lesion epicenter in *Cx3cr1*[creER]*::R26-TdT* mice killed at 1 (**e**), 7 (**f**), and 14 (**g**) days post-injury (dpi) (*n* = 4–5 mice per group/time point). **i** Quantification of the proportional area of spinal cord tissue permeable to FITC-conjugated lectin injected intravenously prior to tissue fixation. **j** Quantification of Ly6G[+] neutrophils at the lesion epicenter at day 1 post-SCI in mice treated with either PLX5622, PLX73086, or the control chow (*n* = 4–5 per group). **k** Quantification of the number of granulo-myelomonocytic cells at the lesion epicenter at 7 and 14 dpi in *Cx3cr1*[creER]*::R26-TdT::LysM*-eGFP mice treated with either PLX5622, PLX73086, or the control diet (*n* = 4–5 mice per group/time point). For all injured mice, treatment was initiated 3 weeks before SCI and continued until sacrifice. Data are expressed as mean ± SEM. \*\**p* < 0.01, \*\*\**p* < 0.001, \*\*\*\**p* < 0.0001, PLX5622 or 1 dpi compared to the control group; and [&&]*p* < 0.01, [&&&&]*p* < 0.0001, PLX5622 compared with the PLX73086 group. Statistical analysis was performed using a one-way (**d–g**, **i**, **j**) or two-way (**k**) ANOVA followed by a Bonferroni's post hoc test. Scale bar: (**a–c**, in **c**) 200 µm

macrophages, including microglia, and depend on colony-stimulating factor 1 receptor (CSF1R) signaling for their survival[16]. Thus, to better understand the role of microglia and MDMs in SCI, we compared the effects of two novel CSF1R inhibitors from Plexxikon: (1) PLX5622, a drug that crosses the blood–brain barrier (BBB) and eradicates nearly all microglia in the brain[9,10,17–19], and (2) PLX73086, a CSF1R inhibitor that does not deplete resident microglia because of its low BBB penetration (Dr. Andrey Rymar, Plexxikon, personal communication). Treatment of uninjured C57BL/6 mice with PLX5622 for 3 weeks resulted in depletion of 97.9 ± 0.6% of spinal cord microglia (Fig. 2a–d and Supplementary Figure 5). In contrast, PLX73086 did not affect the microglial population. To examine the effects of the two CSF1R inhibitors on peripheral immune cells, we next performed a cytometric analysis of leukocyte subsets in the blood, spleen, and bone marrow. No changes were observed after 3 weeks of treatment with the different diets (Supplementary Figure 6a–c). Altogether, these data indicate that PLX5622 can be used to selectively and nearly completely eliminate spinal cord

microglia without significantly affecting peripheral leukocyte counts under steady state in vivo conditions in mice.

To examine the long-term effect of continuous PLX5622 treatment on microglia depletion, we counted the number of microglia per mm[2] at 1, 7, and 14 days post-SCI (Fig. 2e–h). Mice were fed chow containing either PLX5622, PLX73086, or vehicle (without gavage) starting 3 weeks before SCI and until time of sacrifice. Again, treatment of C57BL/6 or *Cx3cr1*[creER]*::R26-TdT* mice with PLX5622 eliminated virtually all microglia at each of the above time points. Although no changes were observed at day 1 and day 14 between PLX73086-treated mice and those fed the control diet (Fig. 2e, g), we found that the total number of TdT[+] microglia at the lesion epicenter was reduced by nearly half at 7 dpi in animals that received PLX73086 (Fig. 2f). This might be due to a temporary breakdown of the BSCB, which allowed PLX73086 to enter the spinal cord parenchyma and to negatively affect microglial cell survival. In accordance, we detected an increased accumulation of fluorescein isothiocyanate (FITC)-conjugated *Lycopersicon esculentum* agglutinin (LEA) lectin from

day 1 to day 7 post-SCI at the lesion site, but not at 14 dpi (Fig. 2i).

At 1 dpi, a slight neutropenia was observed in C57BL/6 mice treated with PLX5622 or PLX73086 compared to those fed with the control diet (Supplementary Figure 6d). However, the number of infiltrating neutrophils at the site of SCI was similar between all groups at this time (Fig. 2j). No changes in blood leukocyte numbers were observed between groups at 7 and 14 dpi (Supplementary Figure 6e and f), except for a slight and transient decrease in the B cell counts at day 7. Treatment of *LysM*-eGFP mice with CSF1R inhibitors highlighted a transient reduction in the number of myeloid cells at the lesion epicenter in PLX5622-treated mice that was overcome by day 14 (Fig. 2k). Treatment with PLX73086 had no impact on the number of neutrophils and LysM+ cells at the lesion site compared to control treatment (Fig. 2j, k). We interpret that the delayed myeloid cell recruitment in the injured spinal cord of PLX5622-treated mice was caused by the absence of microglia rather than a direct effect on peripheral leukocytes. Overall, these results indicate that PLX5622 can be used to eradicate virtually all spinal cord microglia after SCI with minimal direct effects on leukocytes.

We next investigated the role of microglia in functional recovery after SCI. Chow containing either PLX5622, PLX73086, or no drug (control) was given to C57BL/6 mice starting 3 weeks prior to SCI and then maintained for an additional 5 weeks (Fig. 3a). Similar to uninjured mice that received the control diet, microglia-depleted uninjured animals showed no gross locomotor deficits at the beginning of behavioral testing, as illustrated by the perfect BMS scores and subscores at day 0. No difference was found in terms of BMS score between the PLX73086 and control groups at any of the time points analyzed after SCI (Fig. 3b). However, mice depleted in microglia (PLX5622) exhibited impaired locomotor recovery compared to mice treated with PLX73086 or the control diet at 3, 7, 14, 28 and 35 dpi (Fig. 3b). Statistical differences in the BMS subscores between PLX5622-treated mice and animals of the other groups were detected starting from day 7 up to day 35 (Fig. 3c). At 35 dpi, the average subscore of PLX5622-depleted mice was $2.9 \pm 0.6$ compared to $4.5 \pm 0.2$ for PLX73086-treated mice and $5.4 \pm 0.2$ for mice fed the control diet. These results indicate that microglia play an essential role in recovery from SCI.

**The beneficial actions of microglia occur during week 1 post-SCI.** Recent work has established that the microglia-depleted brain repopulates within one week through local proliferation of residual microglia[20]. We reasoned that we could take advantage of this observation to study the time window during which the neuroprotective and/or neurorepair effects of microglia occur after SCI. We first evaluated the time course of repopulation of microglia in the spinal cord (Supplementary Figure 7). *Cx3cr1*[creER]*::R26-TdT* mice were fed ad libitum for one week with the appropriate treatment. At this time, we counted $75.9 \pm 7.9$ TdT+ microglia per mm$^2$ in the thoracic spinal cord of mice fed the control diet compared to $4 \pm 0.6$ in those treated with PLX5622 (94.7% depletion). The dynamic of the microglial repopulation was then assessed by switching the PLX5622 group to the control diet (Supplementary Figure 7a–m). The diet change resulted in a rapid increase in the number of microglia in the uninjured spinal cord, going from $4.0 \pm 0.6$ TdT+ cells per mm$^2$ at day 0 to $15.8 \pm 2.0$ at day 2 and $35.3 \pm 2.8$ at day 3. By day 7 of withdrawal of CSF1R inhibition, the microglial population had completely recovered, with an average count of $99.5 \pm 2.3$ cells per mm$^2$, exceeding by ~30% the microglia numbers counted in control mice (Supplementary Figure 7m). Repopulated TdT+ cells, observed after 7 days of drug withdrawal, had a ramified

morphology and expressed CD11b and Iba1 (Supplementary Figure 7i–l, n), confirming that they are mature microglia. Only few ($2.3 \pm 1.7\%$) of the repopulated CD11b+ Iba1+ cells were TdT[neg] at 7 days post-withdrawal. This indicates that microglia were repopulated from cells in which the *Cx3cr1* gene promoter was active (i.e. TdT+) at the time of tamoxifen treatment, confirming that adult spinal cord microglia are capable of self-renewal. A similar trend was also seen in C57BL/6 mice, in which the total number of P2ry12+ microglia, after a repopulation period of 7 days, exceeded that of untreated mice by 54% ($116.5 \pm 5.4$ P2ry12+ cells/mm$^2$ compared to $75.7 \pm 2.1$ P2ry12+ cells/mm$^2$).

As described above, spinal cord microglia of adult naïve mice are in low proliferative state (Supplementary Figure 7o). However, 2 and 3 days after PLX5622 removal, 93.5% and 87.4% of the TdT+ microglia, respectively, were actively proliferating based on Ki67 expression. Few TdT[neg] cells also proliferated simultaneously to microglia proliferation but at a reduced rate. Multiple immunofluorescence labeling revealed that the non-microglia proliferating cells were mostly of the oligodendrocyte lineage, with $6.1 \pm 0.9$ and $6.7 \pm 1.9$ Olig2+ TdT[neg] Ki67+ cells/mm$^2$ at 2 and 3 days, respectively (Supplementary Figure 7p). GFAP-positive astrocytes, CD206+ perivascular macrophages, CD13+ pericytes, and CD45+ TdT− blood-derived leukocytes accounted for <1% of the Ki67+ cells. Therefore, our results indicate that residual microglia proliferate extensively and can repopulate the entire spinal cord microglial population within 7 days.

We next sought to determine when the beneficial effects of microglia occur after SCI, as this is critical for the development of a therapeutic approach targeting microglia. For that purpose, C57BL/6 mice were fed PLX5622 for 3 weeks, before switching to a control diet at the time of SCI (Fig. 3d), resulting in microglia depletion at the time of injury but not afterwards. No differences were found in behavioral outcomes between groups (Fig. 3e, f). As microglial repopulation requires about a week to be completed after cessation of treatment with the CSF1R inhibitor (Supplementary Figure 7i–m), we next hypothesized that the beneficial effects of microglia might take place during the first week post-SCI. Since injured mice eat significantly less during the first few days, separate groups of mice were force-fed with either PLX5622 or vehicle by gavage from the time of SCI up to day 7 (Fig. 3g), in addition to the ad libitum access to the drug-containing chow. Importantly, PLX5622-gavaged mice showed impaired locomotor recovery on the BMS scale compared to control mice (Fig. 3h, i). In contrast, SCI mice that were started on the PLX5622 diet at day 3 displayed locomotor scores similar to those observed in the PLX73086 and control groups (Fig. 3j–l). Altogether, these data indicate that activated, proliferating microglia are crucial for protecting/repairing the injured spinal cord and that their beneficial effects take place during the first week post-SCI.

**A microglial scar forms at the astrocyte–immune cell interface.** It was recently proposed that cytokines released by activated microglia in response to CNS injury or disease determine whether astrocytes will have neurotoxic or pro-survival effects[21]. Thus, we next investigated whether microglia play an important role in the formation of the astrocytic scar that develops during the subacute/chronic phases of SCI and which can influence axonal regeneration and functional recovery. As neutrophils and MDMs rapidly accumulate in the injured spinal cord and share common markers with microglia, *Cx3cr1*[creER]*::R26-TdT::LysM*-eGFP reporter mice were initially used to perform an immunofluorescence and ultrastructural characterization of glial scar formation over time (Supplementary Figure 8a). We found that TdT+ microglia accumulate

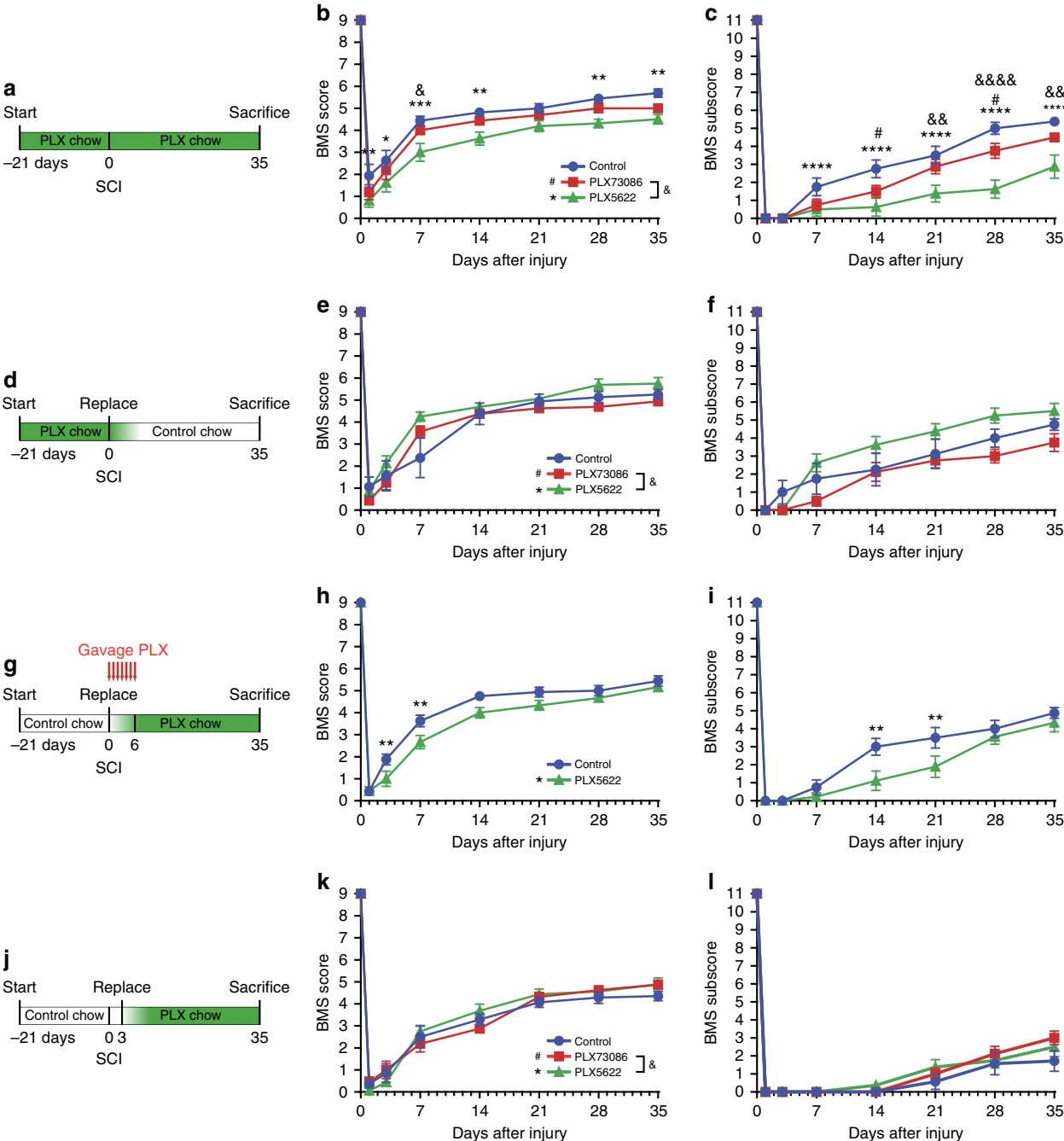

**Fig. 3** Microglia play a key role in recovery of locomotor function during the first week post-SCI. **a**, **d**, **g**, **j** Schematics of experimental design showing the timeline of microglia depletion, spinal cord contusion, behavioral testing using the Basso Mouse Scale (BMS), and sacrifice. CSF1R inhibitors and vehicle were administered in the diet or by oral gavage, as indicated. **b** and **c**, **e** and **f**, **h** and **i**, **k** and **l** Locomotor function was assessed using the BMS score (**b**, **e**, **h**, **k**) and BMS subscore (**c**, **f**, **i**, **l**) over a 35-day period after SCI ($n = 8$ mice per group). Data are expressed as mean ± SEM. *$p < 0.05$, **$p < 0.01$, ***$p < 0.001$, ****$p < 0.0001$, PLX5622 versus the control group; #$p < 0.05$, PLX73086 versus the control group; and &$p < 0.05$, &&$p < 0.01$, &&&&$p < 0.0001$, PLX5622 compared to PLX73086. Statistical analysis was performed using a two-way repeated-measures ANOVA followed by a Bonferroni's post hoc test

mainly around the lesion site, where they make direct contacts with GFAP[+] astrocytes, especially their distal processes, and also with blood-derived LysM[+] cells (Fig. 4a–g, Supplementary Figures 8b–e and 9). This microglial interaction with GFAP[+] astrocytes and blood-derived LysM[+] cells was most apparent starting at 14 dpi and persisted until at least day 35. As described above, nearly half of the microglia detected at the rim of the lesion were actively proliferating at day 7, a response that returned near baseline by day 14 (Fig. 4h). This translated into an increased number of TdT[+] microglia at the lesion epicenter

at 14 dpi (Fig. 4i). However, we noted that the lesion itself was predominantly occupied by LysM[+] neutrophils/MDMs and PDGFR-ß[+] pericytes rather than TdT[+] microglia (Fig. 4j–l). These results indicate that microglia synergize with reactive astrocytes, and perhaps as well with pericytes, to isolate infiltrating immune cells at the core of the lesion. We named this phenomenon "microglial scar" as an analogy to the astroglial-fibrotic scar that develops after SCI and limits the spread of inflammatory cells, and at the same time influences regeneration of the severed axons[22,23].

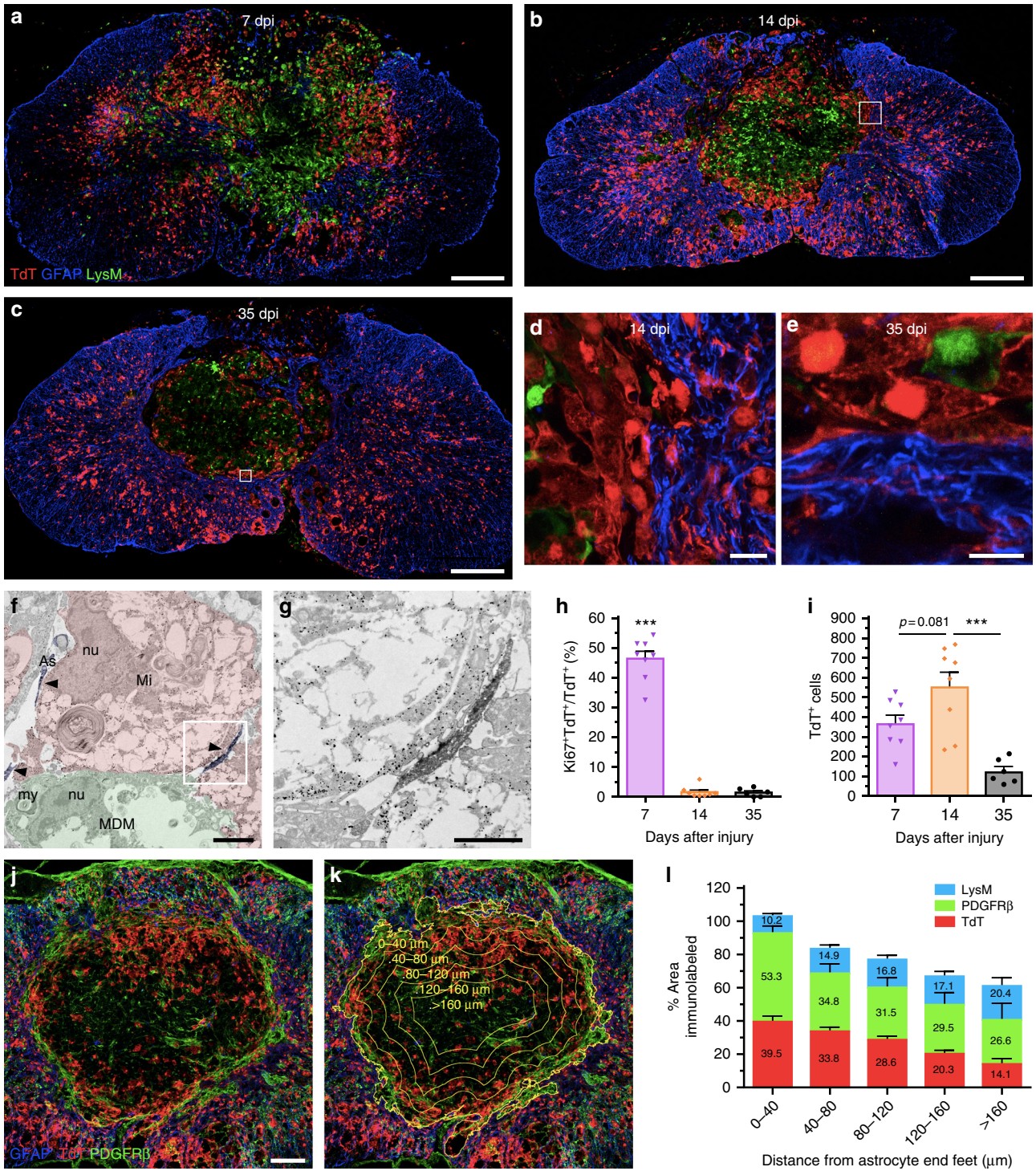

Since a residual fraction of peripheral myeloid cells express TdT in *Cx3cr1*[creER]::*R26-TdT* (Supplementary Figure 1b), we further validated our microglial scar concept in chimeras resulting from the transplantation of β-actin-eGFP bone marrow cells into lethally irradiated *Cx3cr1*[creER]::*R26-TdT* mice (Fig. 5a). Fourteen days after SCI, the microglial scar was still prominent and primarily consisted of TdT+ microglia, with infiltrating bone marrow-derived (eGFP+) cells located at the core of the lesion (Fig. 5b–e). In contrast, very few TdT+ cells were seen at the rim of the lesion when *Cx3cr1*[creER]::*R26-TdT* mice were used as bone

marrow donors for recipient C57BL/6 mice (Fig. 5f–h). Indeed, we found ~25 times less TdT+ cells in *Cx3cr1*[creER]::*R26-TdT* → WT mice (49.5 ± 8.0 cells/mm$^2$) compared to *Cx3cr1*[creER]::*R26-TdT* mice (1204.61 ± 137.8 TdT+ cells/mm$^2$, Fig. 1e, g) at 14 dpi. This suggests that cells from the periphery contribute minimally (<4%) to the total TdT+ cell number in experiments using inducible *Cx3cr1*[creER]::*R26-TdT* mice. Additionally, we inflicted SCI in *Flt3*-cre::*R26-TdT* mice (Supplementary Figure 8f), in which TdT is expressed in HSCs and their progeny (including MDMs and neutrophils), but not microglia[16]. At 14 dpi, a dense

**Fig. 4** A microglial scar forms at the interface between the astrocytic and fibrotic scars. **a–e** Confocal immunofluorescence microscopy of representative spinal cord sections taken at the lesion epicenter at 7 (**a**), 14 (**b**, **d**), and 35 (**c**, **e**) days post-injury (dpi) showing formation of the microglial scar, characterized by the accumulation of TdT$^+$ microglia (red) at the lesion borders, over time. The microglial scar is shown in relation to the infiltration of blood-derived myeloid cells (LysM-eGFP$^+$, green) and formation of the astroglial scar (GFAP-immunoreactive astrocytes, blue). Panels (**d**) and (**e**) are insets of panels (**b**) and (**c**), respectively, showing close-ups of the microglial scar in *Cx3cr1*$^{creER}$::*R26-TdT*::*LysM*-eGFP mice at 14 and 35 dpi. **f** and **g** Immunoelectron microscopy images showing a gold-labeled microglia (Mi) (dense black dots, highlighted in red) located at the lesion border making direct contacts with immunolabeled astrocytic endfeet (As) (diffuse black, highlighted in blue and pointed by arrowheads) and a monocyte-derived macrophage (MDM, highlighted in green). The intimate relationship between the microglia and distal astrocytic processes is shown at high magnification in the inset (**g**). nu = nucleus, my = myelin debris. **h** Percentage of microglia (TdT$^+$) that are actively proliferating (Ki67$^+$ TdT$^+$) at the lesion epicenter at 7, 14, and 35 dpi. **i** Counts of microglia (TdT$^+$) at the lesion epicenter at 7, 14, and 35 dpi. **j** and **k** Confocal images showing the presence of TdT$^+$ microglia (red), PDGFRβ$^+$ pericytes/fibroblasts (green) and GFAP$^+$ astrocytes (blue) at the lesion epicenter at 14 dpi. The distance from astrocyte endfeet is depicted by the yellow lines and indicated (**k**). **l** Percentage area occupied by microglia (TdT$^+$, red bars in the histogram), pericytes/fibroblasts (PDGFRβ, green), and blood-derived myeloid cells (LysM-eGFP$^+$, blue) as a function of distance from astrocyte endfeet ($n = 4$–9 mice). Data are expressed as mean ± SEM. *$p < 0.05$, **$p < 0.01$, ***$p < 0.001$, compared to the other time points. Statistical analysis was performed using a one-way ANOVA followed by a Bonferroni's post hoc test. Scale bars: (**a–c**) 200 μm; (**d** and **e**) 20 μm; (**f**) 5 μm; (**g**) 2 μm; (**j** and **k**) 100 μm

layer of CD11b$^+$ TdT (Flt3)$^{neg}$ cells was clearly detectable at the interface between GFAP$^+$ astrocytic end-feet on the outside of the lesion and CD11b$^+$ TdT (Flt3)$^+$ blood-derived myeloid cells inside the lesion (Supplementary Figure 8g–j). This once again indicates a minimal contribution of blood-derived myeloid cells to the microglial scar that rapidly forms after SCI.

To eliminate the possibility that long-living TdT$^+$ perivascular and/or meningeal macrophages could be at the origin of the microglial scar, immunofluorescence staining was performed to detect the co-expression of TdT and markers of perivascular and meningeal macrophages. CNS border-associated macrophages were defined based on expression of macrophage mannose receptor (CD206) and major histocompatibility complex class II (MHCII)[24,25]. Although some TdT$^+$ microglia at the rim of the lesion were CD206$^+$ at 14 dpi, they weakly expressed CD206 compared to perivascular and meningeal macrophages (Fig. 5i–o and Supplementary Figure 10a–c). We failed to detect MHCII expression on TdT$^+$ microglia (Fig. 5p and Supplementary Figure 10d and e). Instead, the MHCII signal was restricted to TdT$^{neg}$ myeloid cells that infiltrated the lesion core, as well as perivascular and meningeal macrophages. In sum, our results demonstrate that a microglial scar, consisting of primarily proliferating microglia, forms at the border of the lesion after SCI.

**Microglia induce astrocytic scar formation via IGF-1.** Adequate astrocyte reactivity and glial scar formation have been shown to be vital for recovery of neurological functions after SCI[26,27]. Strikingly, we observed that astrocytes located just outside of the lesion core formed a less compact scar when microglia were depleted using PLX5622 compared to the control treatment at 14 dpi (Fig. 6a–d). Notably, GFAP$^+$ astrocytes were oriented randomly and not aligned in any particular direction in PLX5622-treated SCI mice. This disorganized astroglial scar was accompanied by an increased infiltration of blood-derived myeloid cells inside the spinal cord parenchyma, outside of the primary lesion. Since glial scar borders that surround the site of SCI are typically formed by newly proliferated astrocytes[28], we next investigated whether microglial depletion was associated with changes in astrocytic proliferation. To enable accurate counting of pro-liferating (BrdU$^+$) astrocytes, these cells were identified based on expression of Sox9, a nuclear protein exclusively expressed by astrocytes in the adult CNS (except for ependymal cells)[29]. As shown in Fig. 6e–k, the number Sox9$^+$ BrdU$^+$ astrocytes at the lesion epicenter and in adjacent areas was reduced by ~40–55% in PLX5622-treated mice compared to the control group at 7 dpi. This suggests that, after SCI, microglia release molecules that trigger astrocyte proliferation and astrocyte scar formation.

Given the recent demonstration that the astrocyte response is determined by cytokines released by activated microglia in models of neuroinflammation (LPS) and CNS injury (optic nerve crush)[21], we studied the expression of cytokines identified as confirmed or potential inducers of A1 (TNF, IL-1α, IL-1β, IL-6) and A2 (transforming growth factor beta 1 (TGF-β1), IGF-1) phenotypes using in situ hybridization (ISH). From 7 to 14 dpi, the period during which we observed the greatest proliferation of astrocytes and formation of the astroglial scar, mRNA transcripts for the proinflammatory cytokines IL-1α, IL-1β, IL-6, and TNF-α were weakly expressed at the lesion epicenter (Fig. 7a–d). In contrast, we detected strong expression of TGF-β1 and IGF-1 mRNAs at these times (Fig. 7e–l). The spatial distribution of TGF-β1-expressing and IGF-1-expressing cells correlated with the microglial scar, with more ISH signal at the lesion border than at the lesion center. Accordingly, selective depletion of microglia resulted in a decrease of TGF-β1 and IGF-1 mRNA signals. To further demonstrate the involvement of microglia-derived TGF-β1 and IGF-1 in the formation of the astrocytic scar, we treated primary astrocytes with recombinant forms of these proteins. As shown in Fig. 7m, n, treatment with IGF-1, but not TGF-β1, induced the proliferation of astrocytes (BrdU incorporation) and their migration towards the site of injury (scratch assay). Immunolabeling for IGF-1 in tissue sections from *Cx3cr1*$^{creER}$:: *R26-TdT* mice confirmed that scar-forming TdT$^+$ microglia are the principal cellular source of this factor at 7 dpi (Fig. 7o–q). Accordingly, in vivo inhibition of the IGF-1 receptor using an antagonist, OSI-906[30], resulted in reduced expression of the astrocyte-specific marker Sox9 in the injured spinal cord of C57BL/6 mice (Fig. 7r). Thus, microglia-derived IGF-1 triggers astrocyte proliferation and promotes astrocyte scar formation after SCI.

Since reactive astrocytes were previously shown to exert protective functions after SCI[26], we next aimed to determine whether microglial depletion affects tissue damage. Quantification revealed that disrupted glial scar formation in PLX5622-treated mice is correlated with an increase of the lesion core area at 7 days post-SCI (Fig. 8a–c). Although no significant differences were observed between groups regarding the lesion core area at 14 and 35 dpi, we detected the presence of several secondary satellite lesions outside of the primary lesion in the spinal cord parenchyma of microglia-depleted mice (Fig. 8d–i). Secondary satellite lesions were devoid of neuronal elements (NF-H$^+$) and instead filled by blood-derived myeloid cells (CD11b$^+$ LysM$^+$) and PDGFRβ$^+$ pericytes/fibroblasts (Fig. 8j, k). Together, these results indicate that microglia play an important role in the formation of the astroglial scar after SCI, which is at least partly mediated by IGF-1, and that failure to carry out this function

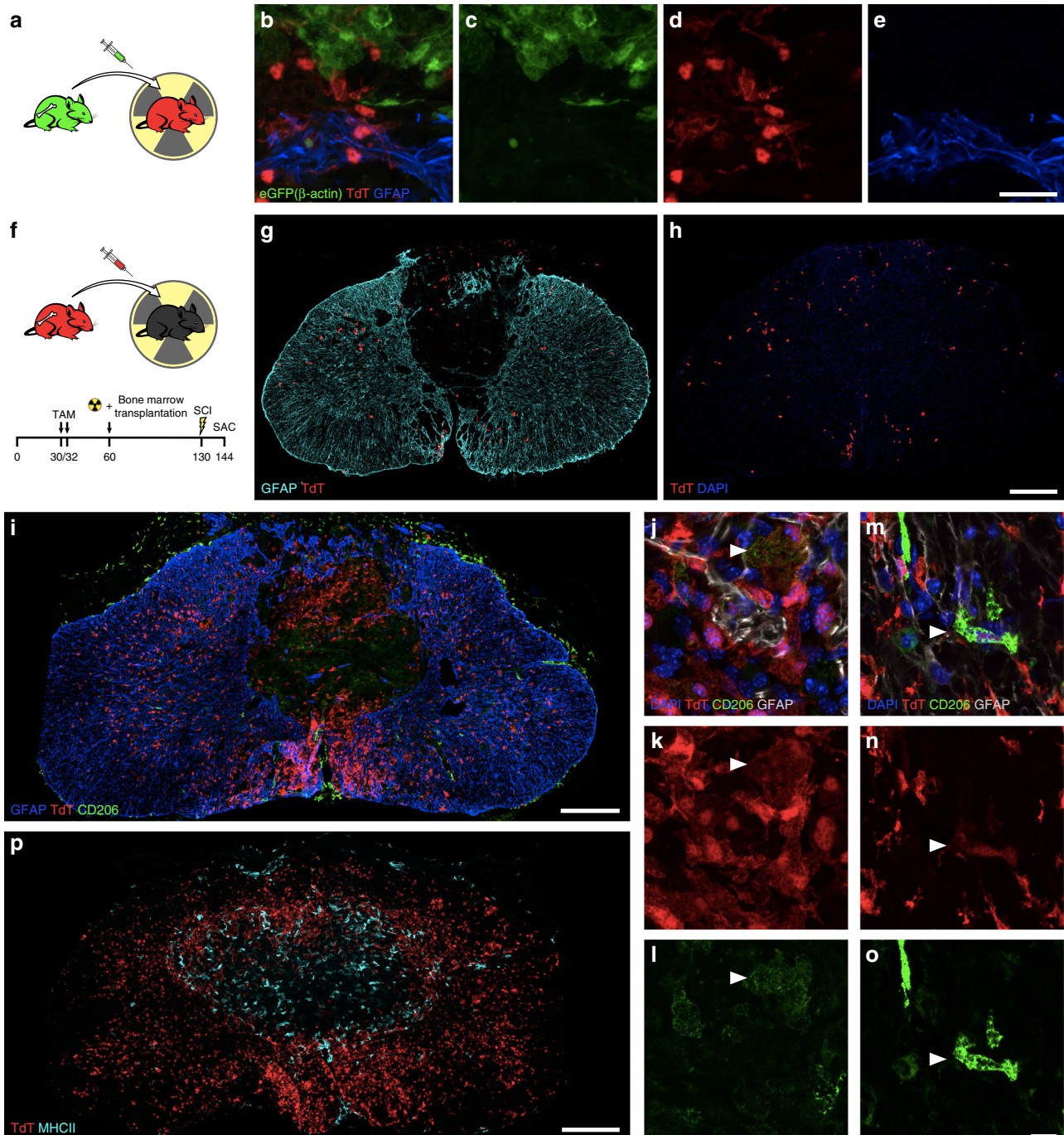

**Fig. 5** The microglial scar is mainly composed of microglia, with few scattered blood-derived myeloid cells and CNS border-associated macrophages. **a** Schematic diagram showing the protocol used to generate radiation bone marrow chimeras in which microglia express TdT and bone marrow-derived cells the GFP reporter. **b–e** Representative confocal images showing the microglial scar formed of TdT+ microglia (red), some of which are in close apposition with GFAP-immunoreactive astrocyte endfeet (blue) on one side and bone marrow-derived cells (eGFP+, green) on the other side at 14 days post-SCI. **f** Schematic of experimental procedure and timeline to generate bone marrow chimeras in which *Cx3cr1*creER::*R26-TdT* mice were used as bone marrow donors for irradiated recipient C57BL/6 mice. **g** and **h** Representative confocal images showing the virtual absence of bone marrow-derived TdT+ cells (red) medial to the astrocytic scar (as defined by GFAP+ astrocyte endfeet in blue), where the microglial scar normally develops, at 14 days post-SCI in *Cx3cr1*creER::*R26-TdT* → WT chimeric mice. **i–o** Confocal images showing the absence (or very weak expression) of CD206 (green) in microglia (TdT+, red) forming the microglial scar at the lesion borders at 14 days post-SCI. In contrast, border-associated macrophages express high levels of the CD206 protein. **p** Representative confocal image showing the absence of colocalization between TdT (red) and MHCII (cyan) in the injured spinal cord of a *Cx3cr1*creER::*R26-TdT* mouse at 14 days. Scale bars: (**b–e**, in **e**) 20 µm, (**g** and **h**, in **h**) 200 µm, (**i**, **p**) 200 µm, (**j–o**, in **o**) 10 µm

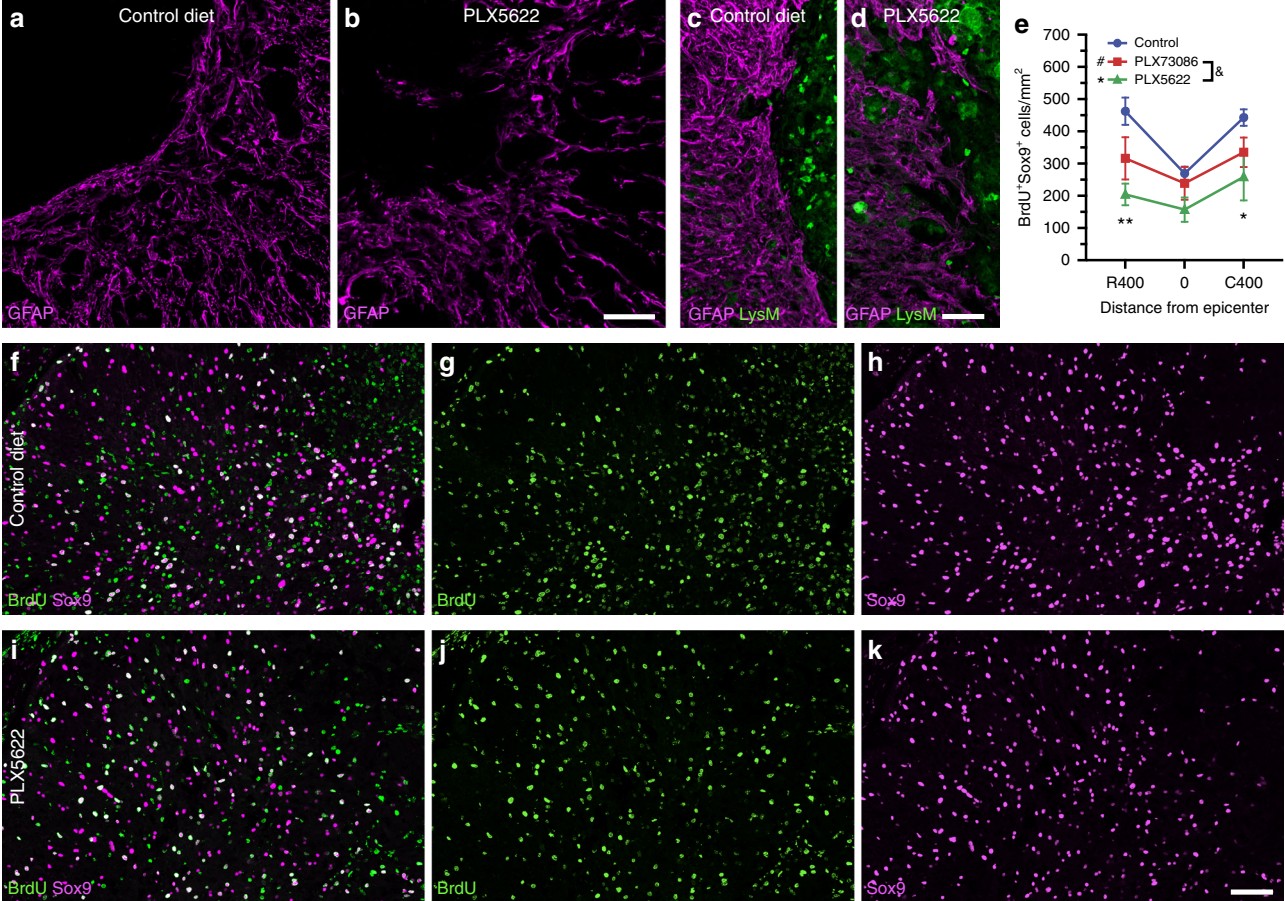

**Fig. 6** The elimination of microglia results in a reduced proliferation of astrocytes and disorganized astrocytic scar at the lesion border. **a–d** Confocal immunofluorescence microscopy of astrocytes (GFAP, purple) in spinal cord sections taken at the lesion epicenter in *Cx3cr1*^creER::*R26-TdT::LysM*-eGFP mice at 14 days post-injury (dpi). In mice fed with the control diet (**a**, **c**), astrocytes adjacent to the site of SCI exhibit elongated processes oriented parallel to the lesion border, thus forming a compact scar. This astrocytic response was compromised in mice depleted of microglia using PLX5622 (**b**, **d**), and associated with clusters of blood-derived myeloid cells (LysM-eGFP$^+$, green cells in **d**) spreading outside of the lesion core. **e** Total counts of Sox9$^+$ BrdU$^+$ cells at the epicenter and both rostral (R) and caudal (C) to the lesion at 7 dpi in mice fed the control diet (blue), PLX73086 (red) or PLX5622 (green) ($n = 4$ mice per group). **f–k** Representative confocal images showing the proliferation of astrocytes (Sox9$^+$, purple cells), as demonstrated by their incorporation of BrdU (green cells), in mice treated with PLX5622 (**f–h**) or the control diet (**i–k**) and killed at 7 dpi. Data are expressed as mean ± SEM. *$p < 0.05$, **$p < 0.01$, PLX5622 versus the control group. Statistical analysis was performed using a two-way ANOVA followed by a Bonferroni's post hoc test. Scale bars: (**a** and **b**, in **b**) 50 μm, (**c** and **d**, in **d**) 50 μm, (**f–k**, in **k**) 50 μm

results in widespread inflammation and the appearance of satellite lesions.

**Microglia prevent death of neurons and oligodendrocytes after SCI.** Having established that the lesion load was increased and functional recovery worse in microglia-depleted mice at 35 days post-SCI, we next determined whether microglial elimination would influence the survival of neurons and oligodendrocytes. No differences in numbers of HuC/HuD$^+$ neurons and Olig2$^+$ CC1$^+$ mature oligodendrocytes were seen between groups in the absence of injury (Fig. 8l, p). However, there were fewer neurons and oligodendrocytes in spinal cord sections spanning the lesion site in the PLX5622 group compared with the other groups at 35 dpi (Fig. 8m–o, q). As expected, this difference resolved at distances > 1.0 mm from the lesion site, thus suggesting that microglia are necessary for the survival of neurons and oligodendrocytes following an insult. Altogether, our results indicate that microglia play a neuroprotective role during SCI.

**Boosting microglial proliferation enhances recovery after SCI.** The above results demonstrate the importance of the microglial

response in scar formation, protection of neural tissue and functional recovery after SCI. We therefore asked whether an increased microglial population would be beneficial on the outcome of SCI. As we observed that 7 days after the end of PLX5622 regimen the microglial population exceeded the one observed in homeostatic conditions (Supplementary Figure 11a–c), we subjected microglia-repopulated C57BL/6 mice to SCI (Supplementary Figure 11d). Although these mice had ~50% more microglia in their spinal cord at the time of injury, they exhibited no motor benefit (Supplementary Figure 11e and f). These results suggest that microglial density at the time of injury is not the limiting factor in recovery.

Since we previously demonstrated that microglia exert their beneficial effects during the first week post-SCI, coinciding with their maximal proliferation rate (Fig. 1m), we hypothesized that artificially increasing their proliferative response right after the injury and continuously during the first week would lead to improvements. Given the importance of CSF1R signaling in microglia development[1], we tested whether M-CSF injection in the cisterna magna would induce microglial proliferation throughout the spinal cord. As shown in Fig. 9a, treatment with M-CSF induced a dose-dependent effect on microglia

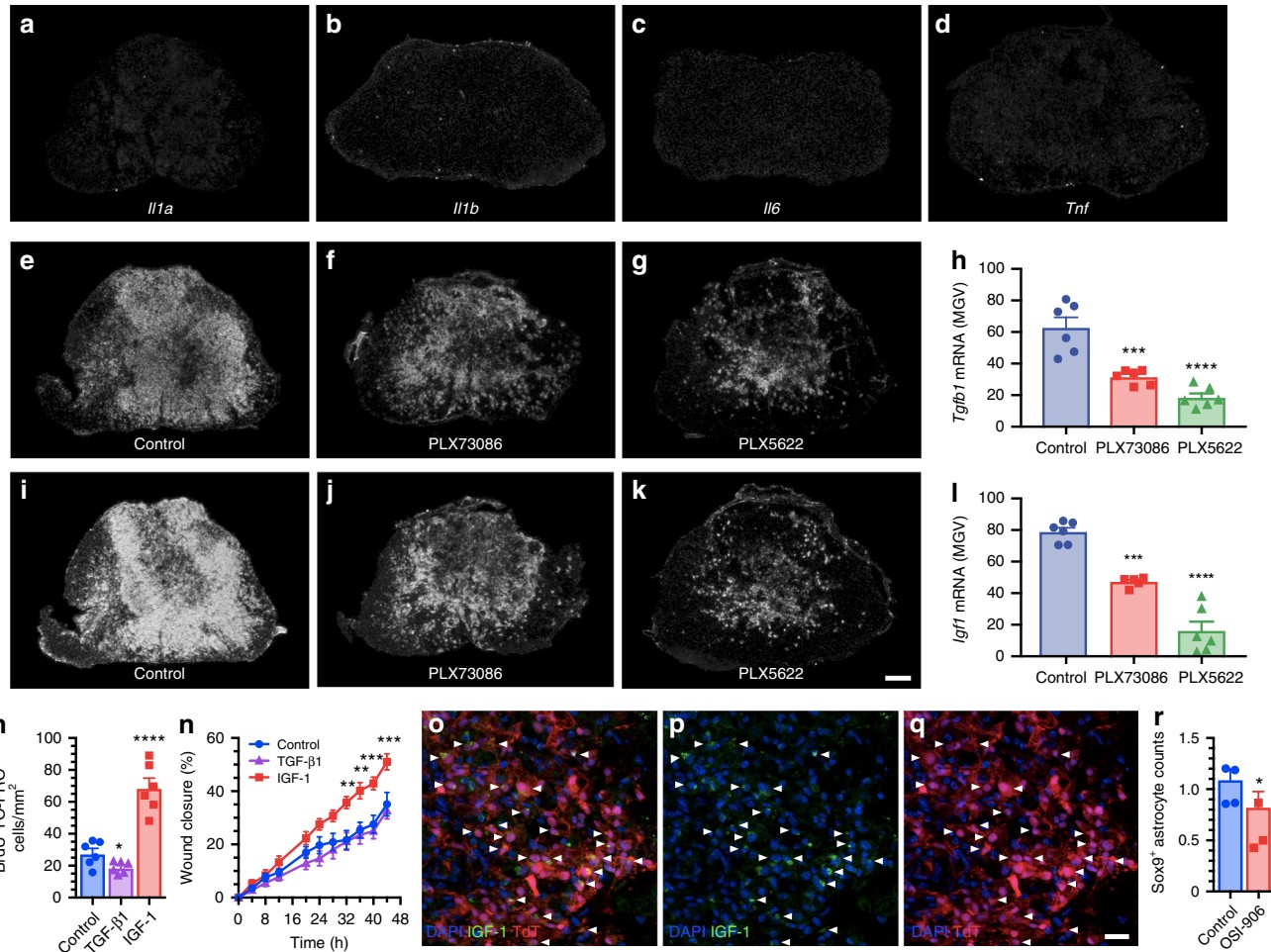

**Fig. 7** Microglia-derived IGF-1 is a potent mitogen for astrocytes and inducer of astrocytic migration towards an injured area. **a–d** In situ hybridization (ISH) signal for the proinflammatory cytokines IL-1α, IL-1β, IL-6, and TNF in the injured mouse spinal cord (lesion epicenter) at 7 days post-SCI (dpi). **e–g**, **i–k** Representative darkfield photomicrographs showing expression of *Tgfb1* and *Igf1* mRNAs at the lesion epicenter at 7 dpi in C57BL/6 mice fed the control diet, PLX73086 or PLX5622. **h**, **l** Quantification of ISH signal (in mean grey values, MGV) for TGF-β1 (**h**) and IGF-1 (**l**) at the lesion epicenter in mice treated with vehicle (Control, blue bars), PLX73086 (red bars) or PLX5622 (green bars) ($n = 6$ per group). **m** Quantification of the number of BrdU$^+$ YO-PRO-1$^+$ nuclear profiles following treatment of primary astrocyte cultures with either TGF-β1, IGF-1 or control solution ($n = 6$ per group). **n** Quantification of the wound closure response in the different groups ($n = 6$ per group). **o–q** Representative immunofluorescence images showing the expression of IGF-1 (green signal, **o**, **p**) by TdT$^+$ microglia (red cells, **o** and **q**) accumulating at the lesion border at 7 dpi. White arrowheads indicate co-localization of IGF-1, TdT, and DAPI (blue). **r** Quantification of Sox9$^+$ astrocytes, expressed as the AUC of the total number of Sox9$^+$ cells (×10$^3$ per mm$^3$) from 800 μm rostral to 800 μm caudal to the epicenter, in the injured spinal cord of C57BL/6 mice treated with the IGF-1R antagonist OSI-906 (red bar) or the vehicle solution (Control, blue bar) ($n = 4$ per group). Data are expressed as mean ± SEM. *$p < 0.05$, **$p < 0.01$, ***$p < 0.001$, ****$p < 0.0001$, compared to the control group. Statistical analysis was performed using either a one-way (**h**, **l**, and **m**) or two-way (**n**) ANOVA followed by a Bonferroni's post hoc test, or a Student's *t*-test (**r**). Scale bars: (**a–g** and **i–k**, in **k**) 200 μm, (**o–q**, in **q**) 20 μm

proliferation, increasing by ~20–25% the number of CD11b$^+$ P2ry12$^+$ microglia in the thoracic spinal cord. Notably, this effect lasted for at least one week. To target the lesion site rather than the entire spinal cord, we incorporated M-CSF into a bioresorbable hydrogel that was injected into the intrathecal space at the site of SCI (Fig. 9b). Previous work has established that the hyaluronan-methyl cellulose hydrogel can provide sustained drug release for 3–7 days[31–33]. Despite the fact that treating *Cx3cr1*$^{creER}$::*R26-TdT* mice with the M-CSF-based hydrogel resulted in a non-significant trend towards a higher number of TdT$^+$ microglia (Fig. 9c), it was sufficient to reduce the lesion area rostral to the epicenter at 7 dpi (Fig. 9d). In addition, the M-CSF-delivering hydrogels improved locomotor recovery from day 7 to day 21 after SCI when compared with PBS-based hydrogels (Fig. 9e, f). This suggests that enhancing the proliferation of microglia limits tissue loss and functional deficits following SCI through regulation of scar tissue formation. Altogether, our

results demonstrate the importance of microglia in protecting the spinal cord after injury.

## Discussion

The role of microglia in SCI has remained obscure for decades. Here, we took advantage of newly developed genetic mouse models, in particular the *Cx3cr1*$^{creER}$ mouse strain, and depletion strategies (e.g. the CSF1R inhibitor PLX5622) that allow us to target microglia specifically to study their role in the context of traumatic SCI. We found that microglia proliferate extensively and accumulate around the site of contusion at 7 days, forming a dense scar at the interface between the fibrotic scar and the yet-to-be-formed astrocytic scar. Notably, the near-complete elimination of spinal cord microglia by PLX5622 treatment led to a reduction in IGF-1 production, a disorganized astroglial scar and the appearance of satellite lesions filled with blood-derived inflammatory cells. This was accompanied by

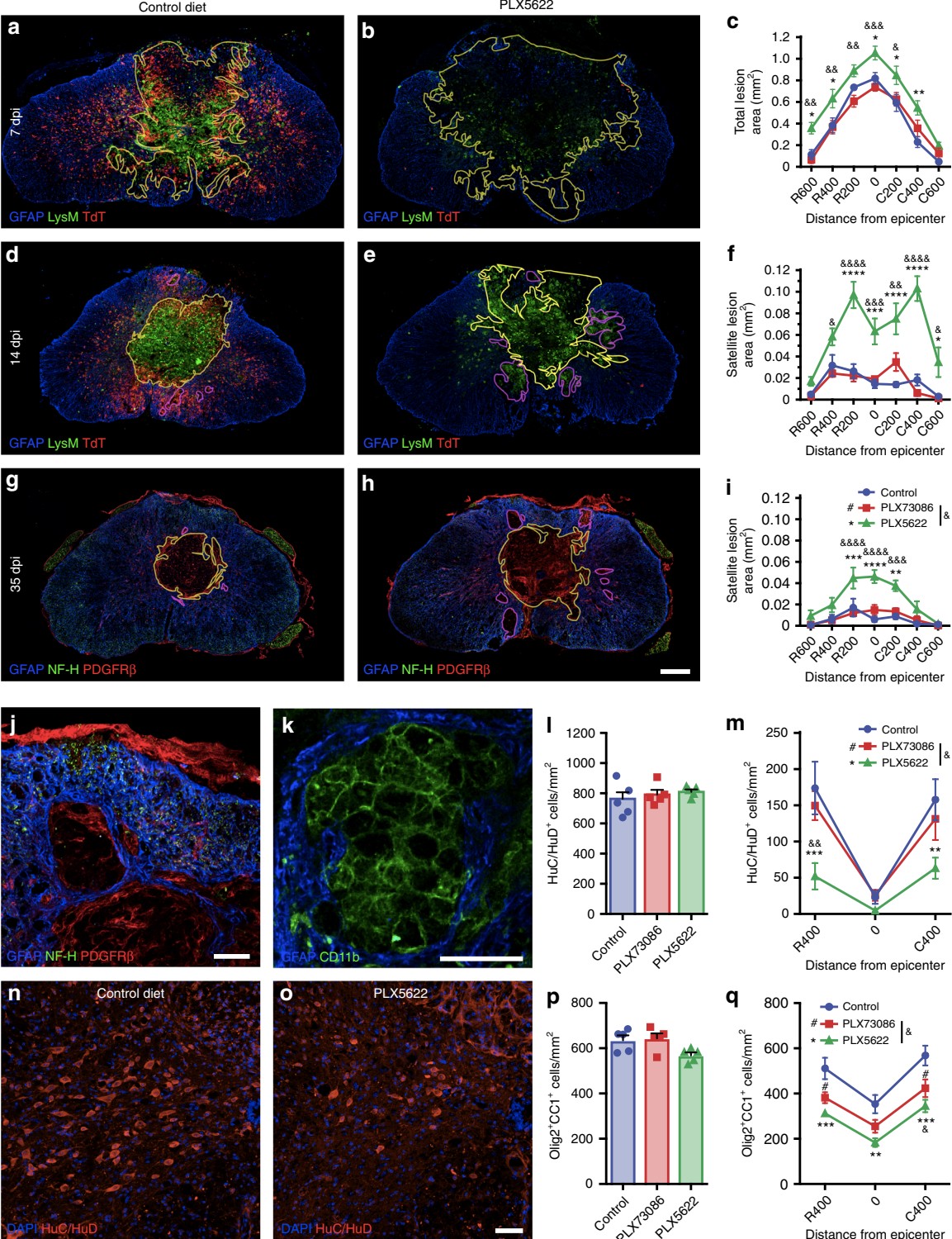

an increased loss of neurons and oligodendrocytes at the site of SCI, as well as greater tissue damage and impairment in locomotor function. The comparison of two CSF1R inhibitors with different BSCB permeability (given at various times before and after SCI) provided strong evidence that the protective function of microglia takes place during the first week post-SCI, matching the peak of proliferation of microglia. Accordingly, local delivery of the microglial proliferation factor, M-CSF, at the site of contusion

significantly improved locomotor recovery compared to vehicle controls.

We initially predicted, based on previous SCI studies using radiation bone marrow chimeras, that MDMs would infiltrate almost exclusively the core of the lesion, while microglia would be found both within spared CNS tissue and the lesion itself[34,35]. However, genetic fate mapping using $Cx3cr1^{creER}::R26\text{-}TdT$ and $Cx3cr1^{creER}::R26\text{-}TdT::LysM\text{-}eGFP$ mouse lines revealed that

**Fig. 8** Microglial depletion results in an increased loss of neurons and oligodendrocytes leading to greater tissue damage after SCI. (**a** and **b**, **d** and **e**, **g** and **h**, **j** and **k**) Confocal immunofluorescence of astrocytes (GFAP+, blue), microglia (TdT+, red cells in **a**–**b**, **d** and **e**), blood-derived myeloid cells (LysM-eGFP+ or CD11b+, green cells in **a** and **b**, **d** and **e**, **k**), neurons/axons (NF-H+, green in **g** and **h**, **j**) and pericytes/fibroblasts (PDGFRβ+, red cells in **g** and **h**, **j**) at the lesion epicenter at 7 (**a** and **b**), 14 (**d** and **e**) and 35 (**g** and **h**) dpi. Yellow and purple lines, respectively, delineate the contours of the primary (core) and satellite lesions, which were surrounded by astrocytic endfeet and characterized by the absence of neuronal elements and presence of cells of non-CNS origin (blood-derived myeloid cells, pericytes, and fibroblasts). Satellite lesions are shown in a microglia-depleted mouse at 35 dpi (**j** and **k**). **c** Quantification of the total lesion area at 7 dpi in mice fed the control diet (blue), PLX73086 (red), or PLX5622 (green) ($n = 5$–6 mice/group). **f**, **i** Quantification of the total area occupied by satellite lesions at 14 (**f**) and 35 (**i**) dpi ($n = 5$–7/group). **l** and **m** Quantification of the number of neurons (HuC/HuD+) in the uninjured spinal cord (**l**), as well as rostral (R) and caudal (C) to the epicenter (**m**), in mice treated with PLX5622, PLX73086, or control at 35 dpi ($n = 5$–8/group). **n** and **o** Representative confocal images taken at the lesion epicenter at 35 dpi immunostained for HuC/HuD (red). DAPI is shown in blue. **p**, **q** Quantification of the number of oligodendrocytes (Olig2+ CC1+) in the uninjured (**p**) and injured (**q**) spinal cord of mice treated with PLX5622, PLX73086 or control at 35 dpi ($n = 5$–8/group). Data are expressed as mean ± SEM. *$p < 0.05$, **$p < 0.01$, ***$p < 0.001$, ****$p < 0.0001$, PLX5622 versus control; #$p < 0.05$, PLX73086 versus control; and &$p < 0.05$, &&$p < 0.01$, &&&$p < 0.001$, &&&&$p < 0.0001$, PLX5622 compared to PLX73086. Statistical analysis was performed using a two-way ANOVA followed by a Bonferroni's post hoc test. Scale bars: (**a** and **b**, **d** and **e**, **g** and **h** in **h**) 200 μm, (**j** and **k**) 50 μm, (**n** and **o**, in **o**) 50 μm

microglia at the site of trauma rapidly die after SCI. Additionally, we found that microglia that surround the lesion site rapidly become activated and proliferate extensively, forming a previously undescribed scar tissue, which we now refer to as the microglial scar. Our findings also contrast with those of Shechter and colleagues who reported that MDMs are restricted to the margins of the lesion and excluded from the center of the lesion[36]. Rather, the margins (borders) of the lesion are entirely occupied by microglia and pericytes/fibroblasts, whereas MDMs are confined to the center of the lesion. This once again shows that, although radiation bone marrow chimeras remain a useful tool, data generated using them must be interpreted with care because: (i) whole-body irradiation harms the blood–CNS barriers and impairs the proliferative capacity of microglia[37], and (ii) HSCs and their progenitors are artificially introduced in the bloodstream as a result of the bone marrow transplant, thus creating a bias towards cells of the hematopoietic compartment[38,39]. Using animal models that did not introduce such artifacts, we can conclude that blood-derived myeloid cells that infiltrate the injured spinal cord remain at the center of the lesion, where they are confined by surrounding tissue consisting of the microglial, fibrotic, and astrocytic scars.

As initially observed in the brain[9], continuous treatment with the CSF1R inhibitor PLX5622 resulted in the depletion of spinal cord microglia (99.6%), that was in our hands more efficient than *Cx3cr1*creER::*R26-iDTR* transgenic mice (77.9%), in which diphtheria toxin (DT) has to be injected to induce cell death[40]. The PLX5622 treatment also avoided the occurrence of the undesired cytokine storm described by Bruttger and colleagues using the iDTR model[37], thus making it a better model to study the role of microglia. We found that the recruitment of MDMs is delayed in the injured spinal cord of PLX5622-treated mice during the acute phase. This finding is consistent with our previous observation that physically injured microglia release damage-associated molecular patterns (DAMPs), such as IL-1α, that initiate sterile neuroinflammation after SCI[41].

Historically, microglial activation in the injured CNS was generally perceived as harmful to both neurons and oligodendrocytes because of the release of high amounts of proinflammatory cytokines, proteases, and reactive oxygen species. Supporting this view is the recent discovery that cytokines derived from activated microglia, such as IL-1α, TNF, and C1q, determine whether astrocytes will have neurotoxic or pro-survival effects in various neurodegenerative disorders[21]. Accordingly, deletion of either of the genes encoding IL-1α, TNF, and C1q in the context of SCI has been associated with an improved locomotor recovery[41–43]. However, the data here show that the elimination of microglia leads to aberrant growth factor production (e.g. IGF-1) and glial scar formation, increased neuronal and oligodendrocyte

death, as well as reduced locomotor performance. In line with our results is a recent stroke study, where microglia depletion using PLX3397, a CSF1R inhibitor that also targets c-KIT and FLT3, increased neuronal death and infarct size in the brain[10]. This once again reinforces the idea that the overall net effect of microglia after CNS injury is neuroprotection. Although the early infiltration of blood-derived myeloid cells at sites of SCI was previously associated with neurotoxicity[44–46], we cannot rule out the possibility that these cells may have contributed to the functional recovery effect seen in PLX5622-treated mice. If it were to be the case, we argue that it would be under the positive influence of microglia as evidence here indicates that the reduction in myeloid cell infiltration was a direct cause of the absence of microglia. In the context of CNS injury, therapies targeting microglia should therefore be aimed at enhancing their neuroprotective function and/or reducing their neurotoxicity rather than complete microglia eradication.

We uncovered that microglia regulate the astrocytic response, in part, through IGF-1. The fact that activated, proliferating microglia are an important source of IGF-1 following a CNS insult is line with findings of Lalancette-Hébert et al. in a mouse model of ischemic stroke[47]. There is also ample in vitro evidence that IGF-1 modulates astrocyte proliferation and the migratory ability of these cells towards a lesion[48,49]. It should be noted, however, that TGF-β1 at the concentration tested in the present study and elsewhere was found not to be mitogenic for astrocytes[50,51]. Still, a role for microglia-derived TGF-β1 in scar formation remains plausible because this cytokine was found to influence astrocytes by acting in synergy with other cytokines and growth factors[51], and as likely to occur in the complex in vivo setting of CNS injury where TGF-β1 neutralization reduces scarring[52–54]. It will therefore be of interest in future work to validate the relevance of these cytokines and their receptors, individually or in combination, in animal models of SCI using cell-specific conditional gene-targeting strategies.

In the absence of microglia, glial scar formation was perturbed, and this resulted in an increased presence of infiltrating blood-derived myeloid cells around the site of trauma. Satellite lesions filled with inflammatory cells have been reported before following depletion of reactive astrocytes[26], as well as in mice with conditional deletion of the *Stat3* gene from astrocytes after SCI[27,55]. Given the importance of astrocytic Stat3 activation in glial scar formation, we speculate that cytokines of the IL-10 families could be additional candidate upstream mediators of these effects (for review, see ref. [56]). Taken together with the finding that microglia-derived cytokines, such as IGF-1, regulate astrocyte function in pathological conditions[21], our results indicate that activated microglia trigger scar formation after SCI.

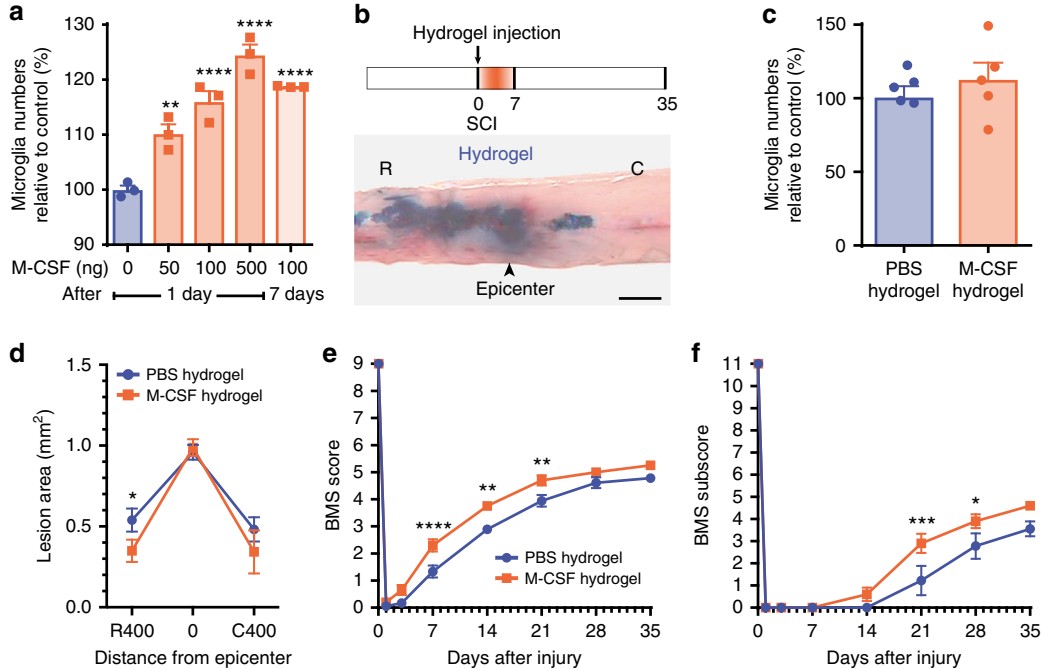

**Fig. 9** Hydrogel delivery of M-CSF at the site of SCI boosted microglial proliferation and enhanced functional recovery. **a** Quantification of the number of microglia (CD11b$^+$ P2ry12$^+$) in the thoracic spinal cord following intra-cisterna magna injection of recombinant murine M-CSF at various doses ($n = 3$ mice per group). **b** Schematic of the experimental design showing the timeline of spinal cord contusion (SCI), hydrogel injection, behavioral testing using the Basso Mouse Scale (BMS), and sacrifice. Below the schematic is a picture showing how much a hydrogel loaded with Evans blue spreads following subdural injection at the site of SCI. **c** Quantification of the number of microglia (TdT$^+$) at the lesion epicenter at 7 days post-injury (dpi) in Cx3cr1$^{\text{creER}}$::R26-TdT mice treated with either M-CSF-based (orange bar) or PBS-based (blue) hydrogels ($n = 5$ mice per group). **d** Quantitative analysis of the total lesion area at the lesion epicenter and both rostral (R) and caudal (C) at 7 dpi in mice treated with M-CSF (orange) or PBS (blue) in hydrogels ($n = 9$–15 mice per group). **e** and **f** Assessment of locomotor recovery using the BMS (**e**) and BMS subscore (**f**) showed that hydrogel delivery of M-CSF increased functional recovery after SCI ($n = 9$–10 mice per group). Data are expressed as mean ± SEM. $*p < 0.05$, $**p < 0.01$, $***p < 0.001$, $****p < 0.0001$, M-CSF-loaded hydrogel compared with PBS-loaded hydrogel. Statistical analysis was performed using a one-way (**a**, **c**), two-way (**d**), or two-way repeated-measures (**e** and **f**) ANOVA followed by a Bonferroni's post hoc test

The identification of the first week post-SCI as the time window for the beneficial effect of microglia suggests that treatments targeting these cells should be initiated promptly. This therapeutic time window is in line with clinical trials that focus on immunomodulators as potential treatments for SCI[57–60]. We also provide evidence that microglia proliferate actively during this period, and that boosting their proliferative response during the early acute phase of SCI using a hydrogel-based M-CSF delivery system further enhances functional recovery.

In light of the current data, we conclude that activated, proliferating microglia play a key role in the formation of the scar that develops after SCI, and that these multicellular interactions serve to sequester blood-derived immune cells in the lesion core, thus protecting non-injured neurons and oligodendrocytes from inflammation-mediated tissue damage.

## Methods

**Animals**. A total of 394 mice were used in this study. C57BL/6N mice were purchased from Charles River Laboratories at 8–10 weeks of age. Cx3cr1$^{\text{creER}}$::Rosa26(R26)-TdT transgenic mice were generated as before[14], and bred with LysM-eGFP knock-in mice created by Dr. T. Graf (Center for Genomic Regulation, Barcelona, Spain) to obtain Cx3cr1$^{\text{creER}}$::R26-TdT::LysM-eGFP. Breeders for Flt3-cre mice were obtained from Drs. Thomas Boehm and Conrad C. Bleul, Max Planck Institute, Freiburg, Germany. All mice were bred in-house at the Animal Research Facility of the Centre de recherche du Centre hospitalier universitaire de Québec–Université Laval. Mice had free access to food and water at all time. All animal procedures were approved by the Centre de recherche du CHU de Québec–Université Laval Animal Care Committee and conducted in compliance with relevant ethical regulations and guidelines of the Canadian Council on Animal Care.

**Tamoxifen treatment**. To induce recombination in Cx3cr1$^{\text{creER}}$::R26-TdT and Cx3cr1$^{\text{creER}}$::R26-TdT::LysM-eGFP mouse lines, mice were treated orally with 10 mg of tamoxifen (dissolved in 1:10 ethanol/corn oil) twice at 2-day intervals starting at postnatal day (P) 30–32. The animals were then allowed a resting period of 28 days prior to SCI to allow sufficient time for the turnover of MDMs and near disappearance of TdT$^+$ cells in the blood, spleen, and bone marrow.

**Spinal cord injury**. Mice were anesthetized with isoflurane and underwent a laminectomy at vertebral level T9–10, which corresponds to spinal segment T10–11. Briefly, the vertebral column was stabilized and a contusion of 50 kdyn was performed using the Infinite Horizon SCI device (Precision Systems & Instrumentation). Overlying muscular layers were then sutured and cutaneous layers stapled. Post-operatively, animals received manual bladder evacuation twice daily to prevent urinary tract infections. Depending on the experiment performed, SCI mice were killed by transcardiac perfusion at 1, 4, 7, 14, and 35 days post-contusion.

**Microglia depletion**. To eliminate microglia, mice were fed PLX5622 (1200 ppm) or PLX73086 (200 ppm) provided by Plexxikon and formulated into AIN-76A chow from Research Diets Inc. For gavage experiments, mice received PLX5622 at 90 mg/kg once a day for 7 consecutive days, starting immediately after SCI. PLX5622 was diluted in 5% DMSO, 0.5% hydroxypropyl methyl cellulose, and 1% polysorbate 80. An equal volume of vehicle was used as control.

**Systemic intravascular lectin injections**. To visualize blood-perfused microvessels and determine the time course and magnitude of BSCB permeability after SCI, mice were injected in the tail vein with FITC-conjugated LEA lectin (100 µg/100 µl, Sigma-Aldrich Canada Ltd.) 10 min prior to transcardial perfusion.

**Bromodeoxyuridine (BrdU) injections**. To label proliferating cells, mice were intraperitoneally injected once daily with BrdU (50 mg/kg of body weight in 0.9% saline) for 6 consecutive days, starting on day 1 after SCI.

**In vivo IGF-1R inhibition**. To determine whether IGF-1/IGF-1R signaling is involved in astrocytic scar formation, mice were orally administered with OSI-906 (also known as Linsitinib, Selleckchem), a CNS-penetrant pharmacological inhibitor of IGF-1R[30]. OSI-906 was formulated daily at 4 mg/ml in 25 mM tartaric acid with shaking and sonication for 15 min and then given by gavage once a day for 7 consecutive days at 40 mg/kg, starting immediately after SCI.

**Intra-cisterna magna (i.c.m.) M-CSF injections**. In the experiment in which we studied the effects of central M-CSF treatment on the proliferation of spinal cord microglia, mice were injected i.c.m. with recombinant murine M-CSF (Pre-proTech) at various doses ranging from 25 to 250 ng/μl in PBS. The i.c.m. treatment consisted of a single injection using a pulled-glass micropipette connected to a 10-μl Hamilton syringe.

**Tissue processing**. For the purpose of histology and immunofluorescence experiments, mice were overdosed with a mixture of ketamine–xylazine and transcardially perfused with PBS followed by 1% PFA, pH 7.4, in PBS. Spinal cords were dissected out and then immersed for 2 days in a PBS solution containing 20% sucrose for cryoprotection. For each animal, a spinal cord segment of 12 mm centered over the lesion site was cut in 7 series of 14 μm-thick coronal sections using a cryostat. For experiments involving ISH, mice were transcardially perfused with a 0.9% saline solution followed by 4% PFA, pH 9.5, in borax buffer. Spinal cords were post-fixed for an additional 2 days in 4% PFA, and then placed overnight in a 4% PFA-borax/10% sucrose solution. Thirty-μm-thick cryostat coronal sections were collected directly onto slides that have a permanent positive charged surface (Leica Biosystems) and stored at –20 °C until ISH was performed.

**Immunofluorescence and confocal imaging**. Immunofluorescence labeling was performed according to our previously published method[61]. Primary antibodies used in this study are of the following sources (catalog numbers in parentheses) and were used at the indicated dilutions: rat anti-BrdU (1:750, Abcam, ab6326), mouse anti-CC1 (1:500, Abcam, ab16794), rat anti-CD11b (1:250, AbD Serotec, MCA711), goat anti-CD13 (1:100, R&D Systems, AF2335), rat anti-CD45 (1:500, BD Biosciences, 553076), rat anti-CD68 (1:2500, AbD Serotec, MCA1957), goat anti-CD206 (1:50, R&D Systems, AF2535), rabbit anti-cleaved caspase-3 (Asp175) (1:250, Cell Signaling Technology, 9661), rat anti-GFAP (1:1000, Invitrogen, 13-0300), rabbit anti-GFAP (1:750, Dako, Z0334), mouse anti-HuC/HuD (1:80, Thermo Fisher Scientific, A-21271), goat anti-iba1 (1:1000, Novus Biologicals, NB100-1028), goat anti-IGF-1 (1:10, R&D Systems, AF791), rabbit anti-Ki67 (1:200, Abcam, ab15580), rat anti-Ly6G (1:2000, BD Biosciences, 551459), chicken anti-neurofilament H (NF-H, 1:500, EMD millipore, AB5539), goat anti-Olig-2 (1:400, R&D Systems, AF-2418), rabbit anti-P2ry12 (1:500, Anaspec, AS-55043A), rabbit anti-PDGFRβ (1:750, Abcam, ab32570), and rabbit anti-Sox9 (1:1000, Millipore, AB5535). For Ki67 immunofluorescence, antigen retrieval was performed using sodium citrate buffer at 95 °C for 5 min. For BrdU, tissue sections were treated with HCl (2.0 N) for 30 min at 37 °C followed by 0.1 M sodium borate (pH 8.5) for 10 min at room temperature. Alexa Fluor secondary antibodies from Thermo Fisher Scientific (1:250 dilution) or Vector Laboratories (1:500) were used for multicolor immunofluorescence imaging, whereas 4′,6-diamidino-2-phenylindole, dilactate (DAPI; 1 μg/ml, Thermo Fisher Scientific) was used for nuclear counterstaining. Sections were imaged on a Zeiss LSM 800 confocal microscope system equipped with 405, 488, 561, and 640 nm lasers. Confocal images were acquired using a Zeiss Axiocam 506 Mono camera and mosaics created using the Zen 2.3 software (Blue edition).

**In situ hybridization**. ISH was carried out to detect mRNAs coding for IL-1α, IL-1ß, IL-6, TNF, TGF-β1, and IGF-1, following our previously published method[62]. Primer pairs and enzymes used for riboprobe synthesis are listed in Supplementary Table 1.

**Quantitative analyses**. For the quantification of microglia (TdT+ or CD11b+ P2yr12+ or CD11b+ TdT+), proliferating microglia (Ki67+ TdT+), neutrophils (Ly6G+), blood-derived myeloid cells (LysM+), astrocytes (Sox9+), neurons (HuC/HuD+), oligodendrocytes (Olig2+ CC1+), proliferating oligodendrocytes (Ki67+ Olig-2+), proliferating astrocytes (BrdU+ Sox9+ or Ki67+ GFAP+), proliferating perivascular macrophages (Ki67+ CD206+), proliferating pericytes (Ki67+ CD13+), and proliferating leukocytes (Ki67+ CD45+ TdT−), the total number of immunolabeled cells per cross section was counted at ×20 magnification using mosaics created from 6 to 12 overlapping confocal images or images obtained using a Zeiss Slide Scanner Axio Scan.Z1. A threshold was applied to the resulting images to trace the contour of the coronal section and a grid of 50 μm × 50 μm positioned over the spinal cord either using ImageJ2 (version 1.51d) or BIOQUANT Life Science software (v. 18.5, Bioquant Image Analysis Corporation). Immunolabeled cells with a DAPI-stained nucleus were then manually counted. Results were presented as the total number of positive cells per cross-section, the average number of positive cells per mm² of tissue section, the percentage of cells that expressed specific markers (Ki67), or the mean area under the curve (AUC) of the number of cells per mm² in a predetermined distance range (Sox9).

For the quantification of the BSCB permeability, the proportional area of tissue stained with the FITC-LEA lectin within the entire coronal section at the lesion epicenter was measured using images taken at ×20 magnification with the Zeiss Slide Scanner Axio Scan.Z1. Thresholding values in Fiji (version 1.52h, National Institutes of Health, NIH) were chosen such that only labeled product resulted in measurable pixels on the digitized images. Contrast between positive signal and background was maximized and held constant between all images. Data were expressed as the proportional area of the tissue section occupied by FITC staining. Proportional area of tissue occupied by GFAP (astrocytes), TdT (microglia), and PDGFRβ (pericytes/fibroblasts) immunolabeling was measured in increments of 40 μm relative to the distance from astrocytic endfeet. The area of tissue occupied by immunostaining in each sampling region was measured using the BIOQUANT Life Science software on video images of tissue sections transmitted by a high-resolution Retiga QICAM fast color 1394 camera (1392 × 1040 pixels, QImaging) installed on a Nikon Eclipse 80i microscope. Thresholding values in BIOQUANT Life Science were chosen such that only immunolabeled product resulted in measurable pixels on the digitized image. Contrast between immunolabeling and background was maximized and held constant between all specimen. Data were presented as the proportional area of the sampling region occupied by immunolabeling.

The calculation of areas of tissue damage after SCI was performed on 1 series of adjacent sections within a predetermined spinal cord segment, including the lesion epicenter and surrounding sections in both directions (i.e., rostral and caudal). Fourteen-μm-thick coronal sections were first immunostained for GFAP as well as LysM, NF-H, and/or PDGFRβ. Sections were then counterstained with DAPI and confocal mosaics prepared as described above. For each section, the outline of the core and satellite lesions were separately traced at ×20 magnification and areas measured using ImageJ2. Both types of lesions were surrounded by GFAP-positive astrocytic processes. However, satellite lesions were adjacent to the lesion core and defined as the absence of normal spinal cord architecture and presence of blood-derived myeloid cells (CD11b+) and pericytes (PDGFRβ+).

The average density of ISH signal was measured within the entire cross-section at the lesion epicenter. Mean grey values (MGV, ranging from 1 to 256 bits) were measured under dark-field illumination on video images of tissue sections transmitted by the Retiga camera, using the BIOQUANT Life Science software. MGV were corrected for the average background signal, which was measured in three boxes of 50 H × 50 W μm (80 H × 80 W pixels) placed in regions where no positive signal was observed. ISH data were expressed as an average MGV of the section.

All quantifications were done blind with respect to the identity of the animals.

**Biological sample collection and processing for cytometry**. Animals were anesthetized with a mixture of 400 mg/kg ketamine and 40 mg/kg xylazine. Blood was collected via cardiac puncture using a 22-gauge syringe and immediately transferred into EDTA-coated microtubes (Sarstedt). Blood samples were then put on slow rotation at room temperature until processing. Prior to collection of the spleen and femurs, animals were transcardially perfused with cold Hanks' balanced salt solution (HBSS) to remove blood from the vasculature.

Spleens were harvested from anesthetized animals and placed in HBSS (without Ca²⁺/Mg²⁺). Spleens were homogenized and passed through a 70-μm nylon mesh strainer (BD Biosciences). Erythrocyte lysis was performed using the ACK buffer. The cell suspension was passed on a second 70-μm nylon mesh strainer and the cell count measured.

For the bone marrow, animals were anesthetized and their left femurs isolated and flushed with HBSS (without Ca²⁺/Mg²⁺) + 2% FBS using a 25-gauge needle. Erythrocytes were lysed using the ACK buffer (NH₄Cl 150 mM, potassium bicarbonate 10 mM, EDTA 0.01 mM). Cells were manually counted with a hemocytometer (Hausser Scientific).

**Flow cytometry**. Cells freshly isolated from the blood, spleen, and bone marrow of SCI mice were analyzed by flow cytometry. In brief, red blood cells were first lysed and the remaining cells incubated with Mouse Fc Block (i.e., purified anti-mouse CD16/CD32; BD Biosciences) for 15 min at 4 ºC to prevent nonspecific binding. Multicolor immunolabeling was then performed for 30 min at 4 °C using the following fluorescently conjugated primary antibodies: PerCP-conjugated anti-CD45 (1:50 dilution), Alexa 700-conjugated anti-CD11b (1:50), BD Horizon™ V450-conjugated anti-Ly6C (1:83), PE-Cy7-conjugated anti-Ly6G (1:50), APC-conjugated anti-CD3e (1:50) and Alexa 488-conjugated anti-B220 (1:50) (all from BD Biosciences; for a full description of these primary antibodies, please refer to our published work[63]). Cells were analyzed using FlowJo software (v. 9.2; Tree Star Inc.) on a FACS LSRII flow cytometer (BD Biosciences). Cells were identified as follows: CD45hi CD11b+ Ly6C+ Ly6G+ cells were considered as neutrophils, CD45hi CD11b+ Ly6Chi Ly6G− cells as Ly6Chigh monocytes (also known as M1 monocytes or monocyte-derived M1 macrophages), CD45hi CD11b+ Ly6Clo Ly6G− cells as Ly6Clow monocytes (or M2 monocytes), CD45hi CD11b− B220+ CD3e− cells as B cells, and CD45hi CD11b− B220− CD3e+ cells as T cells.

**Behavioral analysis**. Recovery of locomotor function after SCI was quantified in an open field using the Basso Mouse Scale (BMS), according to the method developed by Basso and colleagues[64]. All groups of mice exhibited similar

parameters in terms of the impact force and spinal cord tissue displacement prior to BMS testing. All behavioral analyses were done blind with respect to the identity of the animals.

**Immunoelectron microscopy**. Mice were anesthetized with sodium pentobarbital (80 mg/kg, intraperitoneally) and perfused with 3.5% acrolein followed by 4% PFA. Fifty-micrometer-thick coronal sections of the spinal cord were cut in sodium phosphate buffer (50 mM, pH 7.4) using a Leica VT1000S vibratome (Leica Biosystems) and stored at $-20\,^{\circ}C$ in cryoprotectant until use. Spinal cord sections were rinsed in PBS (50 mM, pH 7.4) and then quenched with 0.3% hydrogen peroxide ($H_2O_2$) for 5 min followed by 0.1% sodium borohydride ($NaBH_4$) for 30 min. Afterwards, sections were rinsed three times in PBS and incubated for 1 h in blocking buffer (5% fetal bovine serum, 3% bovine serum albumin, 0.01% Triton X-100) and then overnight with a primary anti-GFAP antibody (1:1000 dilution, Thermo Fisher Scientific). The next day, sections were rinsed three times in PBS and incubated for 2 h with secondary antibody conjugated to biotin (1:500, Vector Laboratories) and for 1 h with Vectastain® Avidin–Biotin Complex Staining kit (Vector Laboratories). Sections were developed in a Tris buffer solution (TBS; 0.05 M, pH 7.4) containing 0.05% diaminobenzidine and 0.015% $H_2O_2$ and then rinsed with PBS and incubated overnight with a primary anti-RFP antibody (1:1000, Rockland). The next day, sections were rinsed in TBS and incubated overnight with secondary antibody conjugated to gold (1.4 nm Nanogold goat anti-rabbit, 1:50, Nanoprobes). Then the sections were washed three times with TBS and twice with 3% sodium acetate. Using the HQ Silver Enhancement kit (Nanoprobes), the staining was revealed at room temperature for 1 min and rinsed quickly with sodium acetate followed by three 5-min washes with PBS. The sections were post-fixed with 1% osmium tetroxide, dehydrated using sequential alcohol baths followed by propylene oxide. Sections were embedded in Durcupan resin (Sigma-Aldrich Canada Ltd.) between ACLAR sheets at 55 °C for 3 days. Ultrathin sections were generated at ~65 nm using a Leica UC7 ultramicrotome. Images were acquired at ×1900 or ×4800 magnification using a FEI Tecnai Spirit G2 transmission electron microscope (Thermo Fisher Scientific) operating at 80 kV and equipped with a Hamamatsu ORCA-HR digital camera (10 MP).

**Production of bone marrow chimeras**. Recipient mice ($Cx3cr1^{creER}::R26-TdT$ or C57BL/6 mice) were exposed to a total body γ-irradiation with a single dose of 7.5 Gy using a cesium-173 source (Gammacell 40 Exactor, MDS Nordion) to destroy HSCs. Recipients were then injected in the tail vein with a total of $9 \times 10^6$ bone marrow cells freshly isolated from either β-actin-GFP or $Cx3cr1^{creER}::R26$-$TdT$ donors, as described before[14]. Briefly, femurs and tibias were harvested from euthanized donor mice and flushed with HBSS (without $Ca^{2+}/Mg^{2+}$) + 2% FBS using a 25-gauge needle. After the bone marrow transplantation, mice were kept in sterile cages and treated for 2 weeks with antibiotics (2.5 ml of Septra (GlaxoSmithKline) in 200 ml of drinking water), and let to recover an additional 8 weeks (for a total of 70 days) before being subjected to SCI.

**Proliferation and scratch wound assays**. Primary cultures of mouse astrocytes were prepared from the cortex of P0–P1 pups, as described in ref. [65], and used from passages 3 to 4. Cells were grown in complete Dulbecco's modified Eagle medium (DMEM) either on glass coverslips coated with poly-L-lysine (0.1 mg/ml) in 24-well plates (for the proliferation assay) or directly into 24-well plates (for the scratch wound assay) at a density of 200,000 cells/well. After 1 day in culture for the proliferation assay and 2 days for scratch wound assay, cells were starved for 18 h after reaching a confluence level of ~70%. The proliferation assay was initiated by the addition of either recombinant mouse (rm) TGF-β1 (50 ng/ml, dissolved in PBS containing 4 mM HCl; R&D Systems), rmIGF-1 (760 ng/ml, dissolved in PBS; R&D Systems) or their respective vehicle. Six hours before the end of the experiment (48 h after growth factor addition), a single dose of 10 μM BrdU (Sigma-Aldrich Canada Ltd.) was added to each well. The BrdU-labeling solution was then removed and cells washed several times with PBS. Cells were next fixed with 4% PFA for 15 min, permeabilized in PBS/0.1% Triton X-100 for 20 min, treated with HCl (2.0 N) for 20 min at 37 °C followed by 0.1 M sodium borate (pH 8.5) for 30 min at room temperature, and then immunostained for BrdU. The total number of proliferating cells was counted on images taken at ×10 magnification, using the "Co-localization and Analyze Particles" plugin in Fiji. Data were expressed as number of BrdU$^+$ YO-PRO-1$^+$ cells per mm$^2$. For the scratch wound assay, confluent cells were starved for 18 h and a linear scratch made in the cell monolayer using a 10-μl sterile pipette tip. Complete DMEM was used to wash the cells three times, after which the growth factors identified above were added in 0.2% serum medium. The closure of the wound was monitored using a Zeiss Axio Observer.Z1 Inverted Microscope equipped with an AxioCam MRm digital camera and an incubation chamber by imaging each well every 4 h over 48 h. The phase-contrast images were analyzed by measuring the closure percentage of the scratch, relative to the initial width ($t = 0$), at various times following exposure to the factors under study.

**Subdural hydrogel implantation**. Hydrogels containing particular M-CSF were prepared as in ref. [31]. Briefly, methylcellulose (310 kDa, Shin-Etsu) and hyaluronan (1200–1900 kDa, FMC) were dissolved in ddH₂O, sterile filtered, lyophilized

(Labconco) under sterile conditions, and stored at 4 °C until use. An initial particulate dispersion was produced by mixing M-CSF powder into 0.5% w/v methylcellulose solution. Drug loaded hydrogels were prepared by physical blending hyaluronan (HA, $1.6 \times 10^6$ g/mol), methylcellulose ($3 \times 10^5$ g/mol), and methylcellulose containing M-CSF in artificial cerebrospinal fluid (aCSF: 148 mM NaCl, 3 mM KCl, 0.8 mM MgCl₂, 1.4 mM CaCl₂, 1.5 mM Na₂HPO₄, 0.2 mM NaH₂PO₄ in ddH₂O, pH adjusted to 7.4, filter sterilized at 0.2 μm) for a final composition of 1.4% w/v HA, 3% w/v MC, and 0.5 μg/ul M-CSF. Components were dispersed in aCSF using a dual asymmetric centrifugal mixer (Flacktek Inc.) and dissolved overnight at 4 °C. On the day of the hydrogel implantation, a 5-μl Hamilton syringe (32-gauge needle with a pre-bent blunt tip) was filled with either M-CSF-loaded or PBS-loaded hydrogels. A needle was then used to gently puncture the dura at the level of the contusion injury and 2 μl of hydrogel injected subdurally immediately following SCI.

**Statistical analysis**. Statistical evaluations were performed with one-way or two-way ANOVA or repeated-measures ANOVA, as indicated in figure legends. Post-ANOVA comparisons were made using the Bonferroni correction. All statistical analyses were performed using the GraphPad Prism software (GraphPad Software Inc.). A $p$-value $< 0.05$ was considered as statistically significant.

**Reporting summary**. Further information on experimental design is available in the Nature Research Reporting Summary linked to this article.

## Data availability
The data that support the findings of this study are available from the corresponding author upon reasonable request.

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

## Acknowledgements

This study was supported by grants from the International Foundation for Research in Paraplegia (P161 to S.L.), the Wings for Life Spinal Cord Research Foundation (WFL-CA-006/11 to S.L.), and the Canadian Institute of Health Research (Foundation Grant to M.S.S.). M.E.J. is supported by the Research Support Program for College Teachers-Researchers of the Fonds de recherche en santé–Québec. We thank Nadia Fortin for her invaluable technical assistance and TransBIOTech–Centre de recherche et de transfert en bio-technologies for giving us access to their inverted fluorescence microscope. We also thank Plexxikon Inc. for providing us with the PLX5622 and PLX73086 compounds, and Drs. Brian L. West and Andrey Rymar for their guidance on how to use these drugs effectively in rodents. We are grateful to Drs. Bleul and Boehm for providing the *Flt3*-Cre mice.

## Author contributions

V.B.-L. conceived the study, designed and performed most of the experiments, analyzed the data, drafted the figures and wrote the manuscript. F.B. designed and performed in vitro experiments, performed immunofluorescence, microscopy imaging, and quantitative ana-lyses, and commented on the manuscript. B.M. performed the in vivo permeability assay and related quantitative analysis, and commented on the manuscript. N.Vallières performed in situ hybridization, immunofluorescence, and flow cytometry experiments, acquired microscopy images, performed quantitative analyses and edited all figures. M.L. generated bone marrow chimeras, performed in vitro and in vivo immunofluorescence, flow cytometry experiments, and quantitative analyses. M.-E.J. performed in vitro assays and related ana-lyses. N.Vernoux performed immunoelectron microscopy. M.-È.T. performed

immunoelectron microscopy and commented on the manuscript. T.F. prepared the hydrogels, trained staff for the subdural implantation and commented on the manuscript. M. S.S. provided the hydrogels and commented on the manuscript. S.L. conceived the study, designed the experiments, supervised the overall project and wrote the manuscript.

## Additional information

**Competing interests:** The authors declare no competing interests.

