## [Peer Review File · Nature Communications]

Reviewers' comments:

Reviewer #1 (Remarks to the Author):

In the manuscript entitled "Microglia are an essential component of the neuroprotective scar that forms after spinal cord injury", the authors evaluate the microglial behavior and their potential role on the recovery from SCI. By taking advantage of the Cx3cr1CreER-fate mapping approach, the authors show that, although microglia can barely be seen within the injury site 1 day after SCI, they are capable of actively proliferate within the first week and their numbers are reestablished. The authors then depleted microglia by using a CSF-1 inhibitor (PLX5622), and showed that microglia-depleted mice present a reduced locomotor recovery after SCI. This observation was correlated with the impaired formation of the astrocytic scar, broader dissemination of peripheral immune cells and increased oligodendrocyte and neuronal death. The manuscript comprises valuable and well-presented data which will contribute to a more in-depth understanding on the role of microglia after SCI. The missing and important part of the study is the link between the different observations, since now the data are merely correlative.

Major points:

- 1- It would help and improve the message of the manuscript if further experiments could be done to present a putative mechanism on how microglia can specifically affect astrocytic scar formation, since the remaining observations (leukocyte tissue distribution, oligodendrocyte/neuronal death) seem to be a consequence of the defected astrocytic barrier. Is this mediated by microglia-related release of harmful cytokines such as TNF α ? The usage of the CX3CR1 ERT2 Cre would give the authors the wonderful chance to delete specific factors just in CX3CR1+ cells in the brain.
- 2- In Fig.1 the authors present the microglial density which is further complemented with the number of Ki67+Td+ proliferating microglia at different timepoints after SCI. As the authors correctly state in their manuscript, it was previously shown by Askew et al. that "microglial population remains stable throughout life by coupled cell death and cell proliferation" during homeostasis. It would be important to show what happens in the context of SCI. The authors show the proliferating microglia but then they seem to assume that absence/reduce microglial numbers are due to cell death and no data is presented to support such a statement. The authors should present the number of Td+ apoptotic microglia (by using TUNEL assay, for example). This would help to explain the almost absolute absence of microglia at day 1 and the decrease of microglial density observed between day 14 and day 35.
- 3- To strengthen the data from Fig.2d-g and the differential capacity of the PLX73086 inhibitor to deplete microglia, the authors should evaluate the leakage of the BBB (with Evans blue for example) at 1, 7 and 14 days after SCI.
- 4- In Suppl. Fig.5 the authors show that cell proliferation is not only observed for microglia but also for other cell types. Among the cells observed to proliferate, it is mentioned that the "unknown cells" are associated with vessels. It would be important to address if these cells are perivascular macrophages and whether CNS macrophages are also affected by the PLX5622.
- 5- How is the astrocytic scar and the distribution of peripheral immune cells in the M-CSF treated mice? Is it the case that the astrocytic barrier is even more cohesive? In the overall, the data presented in this manuscript is really well presented but is still purely correlative and there's no hint on how the microglial elimination or increased proliferation relates with the downstream observations (astrocytic scar formation and neuron/oligodendrocytes number). The authors should at least evaluate the cytokine production by microglia, including the M-CSF experiment, to see if the level of cytokines is having a beneficial effect on astrocytes and consequent astrocytic scar formation.
- 6- Because the CX3CR1 ERT2 Cre lines targets in the CNS not only microglia but also perivascular

and meningeal macrophages (as stated in the text by the authors) the authors have to be more cautious in their conclusions within the text because they can't distinguish between both populations.

Minor points:

1- On line 207 the authors mentioned that "Similar to uninjured mice that received the control diet, microglia-depleted uninjured animals showed no motor deficits". It's not clear where this data is shown in the manuscript. Please add "data not shown" if that's the case.

2- On line 237 the authors mentioned that "A similar trend was also seen in C57BL/6 mice, 237 in which the total number of 238 P2ry12+ microglia after a repopulation period of 7 days exceeded that of untreated mice by 239 54% (116.5 ± 5.4 P2ry12+ cells/mm² compared to 75.7 ± 2.1)". It's not clear where this data is shown in the manuscript.

3- There are major omissions of relevant references in the text. References 1-3 were not the first microglia ontogeny describing papers and should therefore be replaced by Ginhoux et al. Science 2010, Schulz C et al. 2012 and Kierdorf et al. Nat Neurosci. Further; lines 425-429: Mildner A, 2007 Nat Neurosci is more appropriate than Ajami et al. Nat Neurosci 2010 because the latter one focusses more on EAE. As for microglia expansion not only the recent Askew et al. study is relevant but also Tay TL Nat Neurosci 2017 that uses a multicolour approach to monitor microglia expansion.

Reviewer #2 (Remarks to the Author):

This paper reports the effects of selective deletion of microglia on the response to spinal cord injury (SCI) in mice. The authors report that microglial deletion disrupted glial scar formation at the injury site and resulted in the spread of immune cells and increased loss of neurons. The authors conclude that microglia contribute beneficially to wound repair after SCI.

From an overall perspective, the basic findings seem interesting, useful and potentially important. The bulk of the work seems well conducted and reliable. Nevertheless, there are quite a number of details that need to be dealt with. In addition, the authors tend to over-interpret and over-generalize their findings, and the text needs to be revised in this regard.

Specific comments:

1. The cx3cr1 Cre-ER reporter mouse labels not only microglia, but also monocytes and macrophages. The authors here confirm that this is indeed the case although the number of tdT+ leukocytes in the blood is relatively low. Nevertheless, while the number of tdT+/cd11b+ cells in the blood may be low in uninjured mice, the authors should provide an estimate of the percentage of cells that are tdT+ based on their flow cytometer data in the text. Furthermore, the authors should provide information on the degree to which tdT+ cells from the blood can home to the site of injury and proliferate thus making them more prevalent at the site of CNS injury than the concentration in the blood would suggest. Given that the microglia numbers in the lesion core are very low immediately after injury it is possible that cells from the periphery seed and proliferate at the lesion contributing to a substantial proportion of the total tdT+ cell number. The authors could address this unresolved issue by identifying and quantifying the total reduction in tdT+ cell number at the lesion in their bone marrow irradiation/transplantation group. Any reduction in the tdT+ cells in the irradiated group would be attributable to cells coming from the periphery.

2. All bar graphs throughout the manuscript should include an overlay of individual data points. This has become a general requirement/expectation in Nature publications and should be required here.

3. Individual immunohistochemistry channels should be displayed in supplementary figures 3, 5 and 6 where co-localization or differential expression of different markers is being reported.

4. Supplementary figure 5 would benefit from some higher magnification images to show changes in microglia morphology as this is a major claim made in the text.

5. The contents of Supplementary Figure 6 are particularly important to the message of the paper and at least some of the data presented there should be presented a main figure.

6. In the paragraph starting on line 175 the authors should explicitly state the duration of the treatment paradigm that was used for the cell analysis/counts displayed in Figure 2. Was treatment started immediately after injury or before? Was the drug administered in the food or by gavage and food as done for the behavioural study in Figure 3? This information is important because the next figure (figure 3) and text section attributes behavioural differences to particular drug dosing regimens and the reviewer found it difficult to determine the correlation between these behavioural changes and the preceding cell count data. Is the integrity/mechanical compliance of CNS tissue devoid of microglia different to that with microglia? If so, does that mean the application of the contusion injury causes a greater volume of tissue damage and that is responsible for the behavioural deficits? i.e. is the primary injury worse without microglia not simply that the response to the injury is deficient?

7. The authors should include the data being referred to in lines 307-309 in a Supplementary Figure. There is ample space in the Supplementary data section to provide this data and including convincing data is essential to support this important claim. If the authors are not willing to show the data, then they should remove claim from the text.

8. A paragraph in the Discussion (line 444) begins with the claim that "this is the first report that...". Such claims are meaningless and inappropriate, and should be deleted.

9. There are a number of other specific statements in the text that should be modified or deleted:

Lines 217-219. This last sentence of this results paragraph should be deleted from this results paragraph. The concept of 'neuroprotection' is an interpretation for the discussion, not a result, particularly since the "repair mechanisms have yet to be identified".

Lines 299-301. The authors say that they make an analogy to the 'astroglial scar' that forms after SCI and limits the spread of inflammatory cells, but they do not include a literature reference for this point. Given the degree to which they frequently refer to and build on this point, it would seem appropriate to do so.

Lines 345-347. The correlation of a disrupted astroglia border in the microglia depleted tissue does not equal a causation that the microglia are "driving the correct assembly of the astroglial scar after SCI". This is an over-interpretation. Many cellular interactions play a role in this assembly and it is incorrect to say that any one cell type "drives" it. At best the authors can say that the results "indicate that microglia play an important role in astroglial scar formation".

Lines 385 to 392. The authors have attempted to make several conclusions based on data that "did not reach statistical significance. They do so by alluding to trends in the data as a way of supporting these conclusions. This is not at all appropriate and these claims must be removed. Data are either significant or not, and there is no in between. Moreover, these claims are not necessary to try and elevate the paper. The informative biology stands up for itself here making the paper an important contribution. If anything, the manner in which this set of data are presented here diminishes that potential conclusion. In my opinion, this paragraph should be removed. The data presented simply do not support the claim that the hydrogel treatment increased the number of TdT+ cells or reduced lesion area. I strongly encourage the authors to

show the individual data points overlaid on the bar graphs to show the spread of data points. Also, according to the figure legend in Figure 6 the figure d) has an n per group of 3-4 while the behavioural data in figure e, with the same terminal time point, has an n of 9-10 mice per group. Why not show the immunohistochemical data for all animals undergoing behavioural analysis? Why such a disparity between the sample sizes here in the figures? This disparity is troubling.

Lines 493 to 496. The authors end the paper with the sweeping claim that their findings “demonstrate that the role of the microglial scar after SCI is to sequester blood-derived immune cells to the primary lesion core...”. This claim is very much an overstatement and should be appropriately qualified. At best microglial play a role in the multicellular interactions that serve sequester immune cells in the lesion core. The study here presents no evidence that microglial do so on their own, or that this is their primary role after SCI, and the current sentence implies both of these things. The wording also seems very reminiscent of what has been reported by others regarding the astroglia scar. The authors should modify this ending sentence to indicate the multicellular nature of this regulation and cite appropriate references in this regard.

Reviewer #3 (Remarks to the Author):

The manuscript by Bellver-Landete and colleagues describes the microglial cell reaction after a spinal cord injury.

The authors study the still poorly understood reaction of microglia and their subsequent activation after injury. Using CX3CR1-CreERT2/R26-tdT mice, the authors take advantage of the different proliferation rates of CNS microglia and peripheral myeloid cells to selectively label microglial cells in the spinal cord, but not monocyte-derived macrophages. Microglia are highly active, proliferate in the first weeks and generate a ‘microglial scar’. Pharmacological depletion of microglia did not only lead to disrupted scar formation, but also to impaired locomotor recovery.

In general, this study is concisely designed and well written. The results are novel and important. With these observations, the authors provide further insights into the mechanisms of spinal cord injury and its repair. Furthermore, they offer novel routes to develop therapeutic approaches.

Points to be addressed:

1. The immunohistochemical images are hard to appreciate. The authors should display the individual color channels to grey scale and present color views of merged channels (for example Fig. 1a-f (cyan/red) or Suppl. Fig. 5a-c (blue, green, red, cyan)). In addition, the x-axis indicating the distance from the injury center should be labeled by rounded numbers, e.g. 50, 100, 150 μm .
2. In Fig. 2 J, at dpi 7 LysM Egfp+ cells were largely reduced in PLX5622-treated mice. The authors “interpret this to mean that the delayed myeloid cell recruitment in the injured spinal cord of PLX5622-treated mice was caused by the absence of microglia rather than a direct effect on peripheral leukocytes” (Line 198-200). However, the number of tdT+ microglia in PLX73086-treated mice was also reduced by half at dpi 7 (Fig. 2f), while the LysM EGFP+ cells were not altered. These results could be explained as well by the two drugs (PLX5622 and 73086) having different effects on the infiltration of myeloid cells into the lesion site after SCI. Please address.
3. The main conclusion of this study is, that in the first week after SCI, microglia are activated and proliferate drastically to form the so-called ‘microglia scar’ which is pivotal for the functional recovery of the spinal cord. However, this conclusion is mainly based on the cell ablation results using the CSF-1R inhibitor PLX5622. However, in addition to microglia, myeloid cells also express CSF-1R, therefore the functional recovery effect could also be the total outcome of combining the inhibitory effect on microglia and myeloid cells, which is also suggested by the reduction of LysM EGFP+ cells in the first week (also see point 1). The authors claim that the “reduction in myeloid cell infiltration was however transient and overcome by day 14”, that’s exactly the important time window. The authors should provide more precise evidence to distinguish the functional outcome of inhibiting microglia and myeloid cells, respectively.
4. Fig. 3g-i: why are respective experiments with PLX73086 missing?

5. Why are the BMS subscores in Fig. 3I lower than in other experimental conditions?
6. The depletion of microglia (with PLX5622) caused increased oligodendroglial and neuronal cell death after injury. Where are the controls in which healthy mice were treated with PLX5622 and PLX73086?
7. The authors regard day 7 after injury as the most important time point. In the M-CSF hydrogel experiments, however, the MBS scores of the PBS control group (Fig. 6 e and f) are much lower at day 7 than the scores in other control experiments (Fig. 3). Why?
9. Please add the precise time points for the drug treatment (Fig. 2 d-g and also in Suppl. Figures).
11. In line 238 the authors claim that the number of P2ry12+ microglia increase after treatment (>54 %). Where is that dataset? In Suppl.-Fig. 5, no P2ry12+ staining is mentioned.
12. Suppl. Fig. 5 is supposed to show the microglia morphology after repopulation ("Repopulated TdT+ cells seen after 7 days of drug withdrawal had a ramified morphology and expressed CD11b and Iba1 (Suppl. Fig. 5a-c, e)"). However, this is very hard to recognize at the images a to c. Please provide magnified views of exemplary cells. Olig2 is not the perfect marker for the oligodendrocyte lineage in disease models. It might also be upregulated in activated astrocytes. Please use other markers of the oligodendrocyte lineage such as PDGFRalpha (OPC) or CC1 (mature oligodendrocytes)
- 13: In Fig. 6 (line 388), the authors claim that the M-CSF delivering hydrogels strikingly reduce the lesioned area. And yes, the bar in 6d is smaller. But the data do not reach statistical significance. Please rephrase or provide more data.

Minor points:

Please delete in line 354 "counted significantly fewer neurons" (written twice).

Methods part:

Please clarify which mice were used: C57Bl/6N or J?

In summary, the paper shows very interesting new data addressing a unique function of microglia in the spinal cord.

RESPONSES TO THE REVIEWERS'S COMMENTS

We have revised the manuscript to address all the concerns of the Editorial team and outside reviewers.

Altogether, the comments from the Editorial team and the three Reviewers were insightful and incredibly helpful. The suggested modifications have significantly improved the final version of our study by increasing its scientific quality and relevance. Our responses to the Reviewers' comments are cited point by point below. To facilitate their work, all changes are highlighted in yellow in the revised manuscript. We thank the reviewers for their insightful comments and provide the following responses:

Reviewer 1:

In the manuscript entitled “Microglia are an essential component of the neuroprotective scar that forms after spinal cord injury”, the authors evaluate the microglial behavior and their potential role on the recovery from SCI. By taking advantage of the Cx3cr1CreER-fate mapping approach, the authors show that, although microglia can barely be seen within the injury site 1 day after SCI, they are capable of actively proliferate within the first week and their numbers are reestablished. The authors then depleted microglia by using a CSF-1 inhibitor (PLX5622), and showed that microglia-depleted mice present a reduced locomotor recovery after SCI. This observation was correlated with the impaired formation of the astrocytic scar, broader dissemination of peripheral immune cells and increased oligodendrocyte and neuronal death. The manuscript comprises valuable and well-presented data which will contribute to a more in-depth understanding on the role of microglia after SCI. The missing and important part of the study is the link between the different observations, since now the data are merely correlative.

1) It would help and improve the message of the manuscript if further experiments could be done to present a putative mechanism on how microglia can specifically affect astrocytic scar formation, since the remaining observations (leukocyte tissue distribution, oligodendrocyte/neuronal death) seem to be a consequence of the defected astrocytic barrier. Is this mediated by microglia-related release of harmful cytokines such as TNF alpha? The usage of the CX3CR1 ERT2 Cre would give the authors the wonderful chance to delete specific factors just in CX3CR1+ cells in the brain.

The reviewer raises a very good point. We have now performed additional *in vitro* and *in vivo* experiments to elucidate at least part of the mechanism by which microglia-derived factors modulate astrocytic scar formation after SCI, identifying IGF-1 as a key regulator of this process. In particular, we have included in the revised manuscript a new set of data showing the spatial and temporal expression profile of cytokines identified by Liddelow and colleagues (*Nature*, 2017) as confirmed or potential inducers of A1 and A2 phenotypes, that is TNF- α , IL-1 α , IL-1 β , IL-6, TGF- β 1, and IGF-1.

As shown in the **new Fig. 7a-d**, mRNA transcripts coding for proinflammatory cytokines TNF- α , IL-1 α , IL-1 β , and IL-6 are barely detectable in the injured spinal cord at 7-14 days post-SCI (images for the 7-day time point are shown), the period during which we observed a massive proliferation of astrocytes and formation of the astroglial scar (see **new Fig. 6**). In contrast, microglia that accumulate around the site of SCI express high levels of TGF- β 1 and IGF-1 mRNAs (**new Fig. 7e-l**). This was demonstrated by the distribution pattern of TGF- β 1- and IGF-1-expressing cells as a function of time post-SCI, as well as the fact that selective depletion of microglia resulted in a dramatic decrease in hybridization signal for these two cytokines/factors. To explore the involvement of microglia-derived TGF- β 1 and IGF-1 in the formation of the astrocytic scar that develops after SCI, we next investigated whether treatment of primary CNS astrocytes with recombinant forms of these proteins would induce their proliferation and reactivity. As shown in the **new Fig. 7m-n**, treatment with IGF-1, but not with TGF- β 1, induced proliferation and migration of astrocytes following a pattern that is reminiscent of the astrocytic responses observed after SCI. Immunolabeling against IGF-1 in tissue sections from *Cx3cr1^{Cre-ERT2}::Rosa26-TdT* confirmed that scar-forming TdT⁺ microglia are the principal cellular source of this factor after SCI (**new Fig. 7o-q**). Finally, neutralization of the IGF-1 signaling pathways *in vivo* using a selective inhibitor, OSI-906, significantly reduced the number of astrocytes in the proximity of the lesion (**new Fig. 7r**). Altogether, these data demonstrate that microglia-derived IGF-1 triggers astrocyte proliferation and astrocyte scar formation after SCI. Please note that substantial changes were made to the *Results* (pages 15-16), *Discussion* (page 22), *Methods* (pages 25-29 & 33-34), and *Figure Legends* (pages 50-51) sections to reflect these additions.

We point out that TNF- α was one of the many cytokines investigated in our additional experiments, but was found not to be among the microglia-secreted factors that mediate formation of the neuroprotective barrier composed of reactive astrocytes. This finding is in agreement with the recent discovery by Liddel and colleagues (*Nature*, 2017) that TNF- α rather confers a neurotoxic phenotype to astrocytes.

2) In Fig.1 the authors present the microglial density which is further complemented with the number of Ki67+Td+ proliferating microglia at different timepoints after SCI. As the authors correctly state in their manuscript, it was previously shown by Askew et al. that “microglial population remains stable throughout life by coupled cell death and cell proliferation” during homeostasis. It would be important to show what happens in the context of SCI. The authors show the proliferating microglia but then they seem to assume that absence/reduce microglial numbers are due to cell death and no data is presented to support such a statement. The authors should present the number of Td+ apoptotic microglia (by

using TUNEL assay, for example). This would help to explain the almost absolute absence of microglia at day 1 and the decrease of microglial density observed between day 14 and day 35.

The reviewer brings up an interesting set of questions. We have now included in the revised manuscript the results from additional immunohistochemical analyses that confirm that microglial cell death throughout the injured spinal cord occurs, in part, through apoptosis. These results were incorporated into the **modified Fig. 1i-k**, and are now discussed on pages 5-6 of the *Results* section. However, we have been unable to determine whether necrosis or necroptosis, or other modes of cell death (e.g. pyroptosis, ferroptosis), are also involved based on the lack of reagents available to adequately measure these forms of cell death after SCI *in vivo*. For example, antibodies directed against MLKL and RIPK3 both proved to be unspecific in our PFA-fixed tissue sections, giving diffuse immunostaining with no cell bodies evident.

3) To strengthen the data from Fig. 2d-g and the differential capacity of the PLX73086 inhibitor to deplete microglia, the authors should evaluate the leakage of the BBB (with Evans blue for example) at 1, 7 and 14 days after SCI.

PLX5622 and PLX73086 are very different in terms of composition. Unfortunately, because of confidentiality issues, the structural details of the two molecules cannot be divulged by Plexxikon Inc. However, in our personal communications (details of which can be made available on request), Parmveer Singh (Manager of Business Development, Plexxikon Inc.) and Andrey Rymar (Translational Pharmacology Associate) have indicated that there are chemical groups on PLX73086 that make it much less BBB penetrable than PLX5622, a finding confirmed by their internal unpublished studies which showed a major difference between the plasma and brain concentrations of the two drugs. Importantly, these observations are in full agreement with our spinal cord data which showed that treatment of uninjured mice with PLX5622 resulted in depletion of virtually all microglia, compared to no effect for PLX73086 (**Fig. 2a-d**).

To fully address the reviewer's comment on this issue, the permeability of the blood-spinal cord barrier (BSCB) was examined at 1, 7 and 14 days post-SCI in an attempt to explain the microglia-depleting effects of PLX73086 at 7 days, but not later on. Overall, the data show that BSCB permeability to FITC-conjugated lectin, a tracer whose molecular weight is closer to that of PLX73086 when compared to Evans blue, is increased at the lesion epicenter at 1 and 7 dpi compared to uninjured spinal cord tissue. Vascular permeability changes returned to near baseline values by day 14, thus suggesting a transient leakage of the PLX73086 inhibitor. This could explain why PLX73086 affected microglia only at 7 days. The results of this investigation have been included in the **modified Fig. 2i** of

the revised manuscript, and were discussed as follows (pages 8-9, lines 175-182): "...we found that the total number of TdT⁺ microglia at the lesion epicenter was reduced by nearly half at day 7 in animals that received PLX73086 (Fig. 2f). This might be due to a temporary breakdown of the BSCB, which allowed PLX73086 to enter the spinal cord parenchyma and to negatively affect microglial cell survival. In accordance, there was an increased accumulation of fluorescein isothiocyanate (FITC)-conjugated *Lycopersicon esculentum* agglutinin (LEA) lectin from day 1 to day 7 post-SCI at the lesion site, but not at day 14 post-SCI (Fig. 2i)."

4) In Suppl. Fig. 5 the authors show that cell proliferation is not only observed for microglia but also for other cell types. Among the cells observed to proliferate, it is mentioned that the "unknown cells" are associated with vessels. It would be important to address if these cells are perivascular macrophages and whether CNS macrophages are also affected by the PLX5622.

To resolve this issue, we have performed additional immunofluorescence (IF) staining using antibodies directed against the following markers: CD13 (a marker of pericytes), CD45 (a pan-leukocyte marker), and CD206 (a marker of perivascular macrophages). Accordingly, we have added to the **revised Suppl. Fig. 7p** IF results showing that unknown cells which proliferate in the spinal cord after cessation of PLX5622 treatment are not perivascular macrophages (CD206⁺), pericytes (CD13⁺) or blood-derived leukocytes (CD45⁺ TdT⁻). These new data are discussed as follows in the revised manuscript (page 11, lines 245-247): "GFAP-positive astrocytes, CD206⁺ perivascular macrophages, CD13⁺ pericytes, and CD45⁺ TdT⁻ blood-derived leukocytes accounted for less than 1% of the Ki67⁺ cells." The legend of **Suppl. Fig. 7** (SI text file) has also been modified to reflect these changes.

Regarding the statement that "Most Ki67⁺ cells that could not be identified in the thoracic spinal cord after PLX5622 removal were either located in the central canal or associated with blood vessels", it mainly concerned one particular time point (i.e. 7 days post-cessation of treatment). We have therefore deleted this statement and removed all mentions of the "unknown cells" from the manuscript to avoid confusion to the reader.

5) How is the astrocytic scar and the distribution of peripheral immune cells in the M-CSF treated mice? Is it the case that the astrocytic barrier is even more cohesive? In the overall, the data presented in this manuscript is really well presented but is still purely correlative and there's no hint on how the microglial elimination or increased proliferation relates with the downstream observations (astrocytic scar formation and neuron/oligodendrocytes number). The authors should at least evaluate the cytokine production by microglia, including the M-CSF experiment, to see if the level of cytokines is having a beneficial effect on astrocytes and consequent astrocytic scar formation.

As requested by Reviewer 1, we have performed a number of additional experiments to address these concerns and added them to the **new Fig. 7**. First, we have examined the expression of cytokines identified as confirmed or potential inducers of A1 (TNF, IL-1 α , IL-1 β , IL-6) and A2 (TGF- β 1, IGF-1) phenotypes in the injured spinal cord (**new Fig. 7a-l**). Second, we have confirmed that activated microglia forming the scar at the lesion borders after SCI produce high amounts of TGF- β 1 and IGF-1 (**new Fig. 7e-l, o-q**), but that only IGF-1 is capable of stimulating *in vitro* the proliferation of astrocytes and their migration towards the lesion (**new Fig. 7m-n**). Third, we have demonstrated that *in vivo* inhibition of the IGF-1 receptor using a selective antagonist, OSI-906, results in reduced expression of the astrocyte-specific nuclear marker Sox9 in the injured spinal cord of C57BL/6 mice (**Fig. 7r**). Taken together with all other changes made to the paper since the original submission, we believe that these new results have greatly contributed to better define the mechanism of action by which microglia exert their beneficial effects after SCI.

6) Because the CX3CR1-ERT2 Cre line targets in the CNS not only microglia but also perivascular and meningeal macrophages (as stated in the text by the authors) the authors have to be more cautious in their conclusions within the text because they can't distinguish between both populations.

We have tempered our conclusions as follows (page 22, lines 503-506): "Yet the possibility remains that perivascular and meningeal macrophages, which are also targeted by the *Cx3cr1^{creER}::R26-TdT* mouse line, could contribute to the microglial scar."

7) On line 207 the authors mentioned that "Similar to uninjured mice that received the control diet, microglia-depleted uninjured animals showed no motor deficits". It's not clear where this data is shown in the manuscript. Please add "data not shown" if that's the case.

We apologize for the confusion created. We have added the following information to clarify this point (pages 9-10, lines 201-204): "Similar to uninjured mice that received the control diet, microglia-depleted uninjured animals showed no gross locomotor deficits at the beginning of behavioral testing, as illustrated by the perfect BMS scores and subscores at day 0."

8) On line 237 the authors mentioned that "A similar trend was also seen in C57BL/6 mice, in which the total number of P2ry12+ microglia after a repopulation period of 7 days exceeded that of untreated mice by 54% (116.5 \pm 5.4 P2ry12+ cells/mm² compared to 75.7 \pm 2.1)". It's not clear where this data is shown in the manuscript.

The text has been revised to indicate that these data are not shown (page 11, lines 235-238): “A similar trend was also seen in C57BL/6 mice, in which the total number of P2ry12⁺ microglia, after a repopulation period of 7 days, exceeded that of untreated mice by 54% (116.5 ± 5.4 P2ry12⁺ cells/mm² compared to 75.7 ± 2.1 P2ry12⁺ cells/mm²; data not shown).”

9) There are major omissions of relevant references in the text. References 1-3 were not the first microglia ontogeny describing papers and should therefore be replaced by Ginhoux et al. *Science* 2010, Schulz C et al. 2012 and Kierdorf et al. *Nat Neurosci*. Further; lines 425-429: Mildner A, 2007 *Nat Neurosci* is more appropriate than Ajami et al. *Nat Neurosci* 2010 because the latter one focusses more on EAE. As for microglia expansion not only the recent Askew et al. study is relevant but also Tay TL *Nat Neurosci* 2017 that uses a multicolour approach to monitor microglia expansion.

We apologize. The correct references have been added to the revised manuscript as follows:

- Page 3, lines 47-48: “Microglia derive from primitive yolk sac progenitors that arise during embryogenesis^{1,2,3}. They are maintained after birth and into adulthood by self-renewal^{4,5...}”
- Page 20, lines 459-461: “HSCs and their progenitors are artificially introduced in the bloodstream as a result of the bone marrow transplant, thus creating a bias towards cells of the hematopoietic compartment^{39,40}.”

1. Ginhoux F, *et al.* Fate mapping analysis reveals that adult microglia derive from primitive macrophages. *Science* **330**, 841-845 (2010).
2. Schulz C, *et al.* A lineage of myeloid cells independent of Myb and hematopoietic stem cells. *Science* **336**, 86-90 (2012).
3. Kierdorf K, *et al.* Microglia emerge from erythromyeloid precursors via Pu.1- and Irf8-dependent pathways. *Nat Neurosci* **16**, 273-280 (2013).
4. Askew K, *et al.* Coupled Proliferation and Apoptosis Maintain the Rapid Turnover of Microglia in the Adult Brain. *Cell Rep* **18**, 391-405 (2017).
5. Tay TL, *et al.* A new fate mapping system reveals context-dependent random or clonal expansion of microglia. *Nat Neurosci* **20**, 793-803 (2017).
39. Mildner A, *et al.* Microglia in the adult brain arise from Ly-6ChiCCR2⁺ monocytes only under defined host conditions. *Nat Neurosci* **10**, 1544-1553 (2007).
40. Ajami B, Bennett JL, Krieger C, Tetzlaff W, Rossi FM. Local self-renewal can sustain CNS microglia maintenance and function throughout adult life. *Nat Neurosci* **10**, 1538-1543 (2007).

Reviewer 2:

This paper reports the effects of selective deletion of microglia on the response to spinal cord injury (SCI) in mice. The authors report that microglial deletion disrupted glial scar formation at the injury site and resulted in the spread of immune cells and increased loss of neurons. The authors conclude that

microglia contribute beneficially to wound repair after SCI. From an overall perspective, the basic findings seem interesting, useful and potentially important. The bulk of the work seems well conducted and reliable. Nevertheless, there are quite a number of details that need to be dealt with. In addition, the authors tend to over-interpret and over-generalize their findings, and the text needs to be revised in this regard.

1) The *Cx3cr1* Cre-ER reporter mouse labels not only microglia, but also monocytes and macrophages. The authors here confirm that this is indeed the case although the number of tdT⁺ leukocytes in the blood is relatively low. Nevertheless, while the number of tdT⁺/cd11b⁺ cells in the blood may be low in uninjured mice, the authors should provide an estimate of the percentage of cells that are tdT⁺ based on their flow cytometer data in the text. Furthermore, the authors should provide information on the degree to which tdT⁺ cells from the blood can home to the site of injury and proliferate thus making them more prevalent at the site of CNS injury than the concentration in the blood would suggest. Given that the microglia numbers in the lesion core are very low immediately after injury it is possible that cells from the periphery seed and proliferate at the lesion contributing to a substantial proportion of the total tdT⁺ cell number. The authors could address this unresolved issue by identifying and quantifying the total reduction in tdT⁺ cell number at the lesion in their bone marrow irradiation/transplantation group. Any reduction in the tdT⁺ cells in the irradiated group would be attributable to cells coming from the periphery.

The reviewer's point is more than valid. To fully address this issue, we have now clearly stated in the revised text the average percentage of CD11b⁺ cells that are TdT⁺ in various tissues of uninjured mice, which resulted in the following modification (page 5, lines 88-90): "In contrast, only a few CD11b⁺ cells in the blood, spleen and bone marrow were TdT⁺, with average colocalization percentages of 3.8 ± 1.7 %, 6.7 ± 1.6 % and 2.4 ± 0.2 %, respectively (Suppl. Fig. 1d-f)." In addition, we also counted the number of TdT⁺ cells at the site of SCI in chimeric mice generated by transplantation of bone marrow cells from *Cx3cr1*^{creER::R26-TdT} mice into wild-type (WT) recipient mice. As illustrated in the **new Fig. 5**, we counted on average 49.5 ± 8.0 TdT⁺ cells at the lesion epicenter in the *Cx3cr1*^{creER::R26-TdT} → WT group at 14 days post-injury, a number approximately 25 times smaller than the average number of TdT⁺ cells in *Cx3cr1*^{creER::R26-TdT} mice at the same time post-injury (1204.61 ± 137.8 TdT⁺ cells/mm²). Despite the fact that there is a definitive bias favoring the recruitment of cells of the hematopoietic compartment in radiation bone marrow chimeras, as discussed on page 20 of the *Discussion* section, these results suggest that cells from the periphery contribute to less than 4% of the total TdT⁺ cell number in our experiments using inducible *Cx3cr1*^{creER::R26-TdT} mice. Perhaps of even greater importance is the fact that only a few of the TdT⁺ cells detected at the site of SCI in

Cx3cr1^{creER}::R26-TdT → WT mice were seen at the rim of the lesion (see **new Fig. 5g-h**), suggesting once more that the microglial scar is mainly composed of microglia. This new information is available on pages 13-14 (lines 298-304): “In stark contrast, very few TdT⁺ cells were seen at the rim of the lesion when *Cx3cr1^{creER}::R26-TdT* mice were used as bone marrow donors for recipient C57BL/6 mice (Fig. 5f-h). We found approximately 25 times less TdT⁺ cells in *Cx3cr1^{creER}::R26-TdT* → WT mice (49.5 ± 8.0 cells/mm², data not shown) compared to *Cx3cr1^{creER}::R26-TdT* mice (1204.61 ± 137.8 TdT⁺ cells/mm², Fig. 1 e, g) at 14 days post-SCI. This suggests that cells from the periphery contribute minimally (less than 4%) to the total TdT⁺ cell number in experiments using inducible *Cx3cr1^{creER}::R26-TdT* mice.” To summarize, we provide additional evidence proving that blood-derived myeloid cells and CNS border-associated macrophages contribute minimally, if at all, to the formation of the microglial scar through multiple approaches, including radiation bone marrow chimeras (**new Fig. 5a-h**), confocal immunofluorescence microscopy (**new Fig. 5i-p & Suppl. Fig. 10**), and fate-mapping studies in two transgenic mouse lines (**modified Fig. 4 & Suppl. Fig. 8-9**).

Finally, we have modified many of our statements and conclusions in the manuscript to avoid over-interpreting our results, as follows:

- Page 3, lines 65-66: “Here, we took advantage of *Cx3cr1^{creER}* mice¹³, a mouse line that allows with an adequate regimen of tamoxifen to label microglia while excluding nearly all MDMs.”
- Page 22, lines 503-506: “Yet the possibility remains that perivascular and meningeal macrophages, which are also targeted by the *Cx3cr1^{creER}::R26-TdT* mouse line, could contribute to the microglial scar. However, this is unlikely as we were unable to colocalize TdT with markers of CNS border-associated macrophages.”
- Title of the **new Fig. 5**: “The microglial scar is mainly composed of microglia, with few scattered blood-derived myeloid cells and CNS border-associated macrophages.”

2) All bar graphs throughout the manuscript should include an overlay of individual data points. This has become a general requirement/expectation in Nature publications and should be required here.

The bar graphs in **Figs. 2, 4, 6, 7, 8 & 9** and **Suppl. Figs. 1, 7 & 11** have been modified to show the overlay of the individual data points.

3) Individual immunohistochemistry channels should be displayed in supplementary figures 3, 5 and 6 where co-localization or differential expression of different markers is being reported.

Done. Please see **revised Figs. 1 & 5** and **Suppl. Figs. 2, 3, 4, 5, 7, 8, & 10**.

4) Supplementary figure 5 would benefit from some higher magnification images to show changes in microglia morphology as this is a major claim made in the text.

Higher magnification images are now provided in the **modified Suppl. Fig. 7** (Suppl. Fig. 5 in the original submission).

5) The contents of Supplementary Figure 6 are particularly important to the message of the paper and at least some of the data presented there should be presented a main figure.

We agree. A large part of the data presented in Suppl. Fig. 6 in the original submission have now been moved to a main figure, **new Fig. 5**, given their importance in addressing the issue of the *Cx3cr1^{creER}* line specificity. Accordingly, the **new Fig. 5** now presents data from the experiments in which we assessed the contribution of blood-derived myeloid cells, using radiation bone marrow chimeras (**Fig. 5a-h**), and CNS border-associated macrophages (**Fig. 5i-p**) to the formation of the microglial scar. The text was modified accordingly to reflect these changes (see pages 13-14).

6) In the paragraph starting on line 175 the authors should explicitly state the duration of the treatment paradigm that was used for the cell analysis/counts displayed in Figure 2. Was treatment started immediately after injury or before? Was the drug administered in the food or by gavage and food as done for the behavioural study in Figure 3? This information is important because the next figure (figure 3) and text section attributes behavioural differences to particular drug dosing regimens and the reviewer found it difficult to determine the correlation between these behavioural changes and the preceding cell count data. Is the integrity/mechanical compliance of CNS tissue devoid of microglia different to that with microglia? If so, does that mean the application of the contusion injury causes a greater volume of tissue damage and that is responsible for the behavioural deficits? i.e. is the primary injury worse without microglia not simply that the response to the injury is deficient?

As suggested, the treatment paradigm (route of administration, duration, etc.) is now explicitly stated in the *Results* section as well as in the legend of Figure 2. The text was modified as follows to reflect these modifications: 1) Page 8, lines 171-172: “Mice were fed chow containing either PLX5622, PLX73086 or vehicle (without gavage) starting 3 weeks before SCI and until time of sacrifice.”, and 2) Page 47, lines 1068-1069: “For all injured mice, treatment was initiated 3 weeks before SCI and continued until sacrifice.”

Regarding the reviewer’s comment about the possibility that the mechanical compliance of the tissue could have been altered in the absence of microglia, thus resulting in larger lesions after SCI, we argue that the measurement of the spinal cord tissue displacement using the Infinite Horizon (IH) SCI device is a direct assessment of tissue integrity. As described in their *J Neurotrauma* paper (2003) reporting the creation of the IH device, Scheff and colleagues define “tissue displacement” as the distance travelled by the impactor tip from the initial point of contact with the exposed spinal cord to the point at which the predetermined force is attained. Tissue engineering experts agree that one of the most common ways to measure the biochemical properties of the spinal cord is by measuring displacement of the impactor tip after compression, a test referred to as compression testing (Bartlett et al., *Regen Med*, 2016). Importantly, the tissue displacement measure is also considered to be the single most important parameter for generating precise, reproducible experimental contusion SCI (Stokes, *J Neurotrauma*, 1992; Jakeman et al., *J Neurotrauma*, 2000; Grill, *Exp Neurol*, 2005). In the present study, we found no significant difference in the amount of tissue displacement between the three groups (see graph).

7) The authors should include the data being referred to in lines 307-309 in a Supplementary Figure. There is ample space in the Supplementary data section to provide this data and including convincing data is essential to support this important claim. If the authors are not willing to show the data, then they should remove claim from the text.

The requested data have now been included in the **new Fig. 5**. The text has been modified as follows to reflect this addition (pages 13-14, lines 298-304): “In stark contrast, very few TdT⁺ cells were seen at the rim of the lesion when *Cx3cr1^{creER}::R26-TdT* mice were used as bone marrow donors for recipient C57BL/6 mice (Fig. 5f-h). We found approximately 25 times less TdT⁺ cells in *Cx3cr1^{creER}::R26-TdT* → WT mice (49.5 ± 8.0 cells/mm², data not shown) compared to *Cx3cr1^{creER}::R26-TdT* mice (1204.61 ± 137.8 TdT⁺ cells/mm², Fig. 1 e, g) at 14 days post-SCI. This suggests that cells from the periphery contribute minimally (less than 4%) to the total TdT⁺ cell number in experiments using inducible *Cx3cr1^{creER}::R26-TdT* mice.”

8) A paragraph in the Discussion (line 444) begins with the claim that “this is the first report that...”. Such claims are meaningless and inappropriate, and should be deleted.

We apologize. The text has been modified as follows (page 21, lines 478-479): “In this study, we addressed the role of microglia in SCI, highlighting a neuroprotective role for these cells during the acute phase of SCI.”

9) Lines 217-219. This last sentence of this results paragraph should be deleted from this results paragraph. The concept of ‘neuroprotection’ is an interpretation for the discussion, not a result, particularly since the “repair mechanisms have yet to be identified”.

The sentence has been modified as follows (page 10, lines 211-212): “These results indicate that microglia play an essential role in recovery from SCI.”

10) Lines 299-301. The authors say that they make an analogy to the ‘astroglial scar’ that forms after SCI and limits the spread of inflammatory cells, but they do not include a literature reference for this point. Given the degree to which they frequently refer to and build on this point, it would seem appropriate to do so.

The following two reviews are now cited as key supporting references in the revised manuscript (page 13, lines 289-292): “We named this phenomenon “microglial scar” as an analogy to the astroglial-fibrotic scar that develops after SCI and limits the spread of inflammatory cells, and at the same time influences regeneration of the severed axons^{23, 24}.”

23. Sofroniew MV. Astrocyte barriers to neurotoxic inflammation. *Nat Rev Neurosci* **16**, 249-263 (2015).

24. Cregg JM, Depaul MA, Filous AR, Lang BT, Tran A, Silver J. Functional regeneration beyond the glial scar. *Exp Neurol* **253C**, 197-207 (2014).

11) Lines 345-347. The correlation of a disrupted astroglia border in the microglia depleted tissue does not equal a causation that the microglia are “driving the correct assembly of the astroglial scar after SCI”. This is an over-interpretation. Many cellular interactions play a role in this assembly and it is incorrect to say that any one cell type “drives” it. At best the authors can say that the results “indicate that microglia play an important role in astroglial scar formation”.

The text has been modified as suggested (page 17, lines 380-382): “Together, our results indicate that microglia play an important role in the formation of the astroglial scar after SCI, which is at least partly mediated by IGF-1, and that failure to carry out this function results in widespread inflammation and the appearance of satellite lesions.”

12) Lines 385 to 392. The authors have attempted to make several conclusions based on data that “did not reach statistical significance. They do so by alluding to trends in the data as a way of supporting these conclusions. This is not at all appropriate and these claims must be removed. Data are either significant or not, and there is no in between. Moreover, these claims are not necessary to try and elevate the paper. The informative biology stands up for itself here making the paper an important contribution. If anything, the manner in which this set of data are presented here diminishes that potential conclusion. In my opinion, this paragraph should be removed. The data presented simply do not support the claim that the hydrogel treatment increased the number of TdT⁺ cells or reduced lesion area. I strongly encourage the authors to show the individual data points overlaid on the bar graphs to show the spread of data points. Also, according to the figure legend in Figure 6 the figure d) has an n per group of 3-4 while the behavioural data in figure e, with the same terminal time point, has an n of 9-10 mice per group. Why not show the immunohistochemical data for all animals undergoing behavioural analysis? Why such a disparity between the sample sizes here in the figures? This disparity is troubling.

The reviewer raised several important concerns that we have addressed as follows:

- 1) In agreement with Reviewer 2, the following statement in the original submission “...we found that treatment of *Cx3cr1^{creER}::R26-TdT* mice with the M-CSF-based hydrogel increased the number of TdT⁺ microglia at the lesion epicenter at day 7...” has been modified as follows to reflect the non-significance of the data (page 18, lines 418-420): “...treating *Cx3cr1^{creER}::R26-TdT* mice with the M-CSF-based hydrogel resulted in a non-significant trend towards a higher number of TdT⁺ microglia at the lesion epicenter (Fig. 9c)...”.
- 2) The total lesion area presented in the original Fig. 6d was quantified using tissue collected from *Cx3cr1^{creER}::R26-TdT* mice treated with either M-CSF-based or PBS-based hydrogels and killed at 7 days post-SCI, rather than using tissue from animals used for the behavioural assessments. In fact, tissue from hydrogel-treated C57BL/6 mice that underwent behavioural testing was not collected because we failed to detect an effect at 35 days post-SCI/treatment in these animals. We apologize for the confusion.
- 3) To clarify whether hydrogel delivery of M-CSF at the site of SCI reduces or not tissue damage, we generated new tissue material using C57BL/6 mice killed at 7 dpi, as we only had a small number of *Cx3cr1^{creER}::R26-TdT* mice left (4 mice total, which we used to increase the *n* in Fig. 9c reporting the count of microglia). As shown in the **modified Fig. 9d**, the M-CSF-releasing hydrogels significantly reduced the lesion area rostral to the injury epicenter at 7 dpi.

- 4) Individual data points are now presented in all bar graphs, including the **modified Fig. 9** (Fig. 6 in the original submission; see also changes made to **Figs. 2, 4, 6, 7 & 8** and **Suppl. Figs. 1, 7 & 11**).

13) Lines 493 to 496. The authors end the paper with the sweeping claim that their findings “demonstrate that the role of the microglial scar after SCI is to sequester blood-derived immune cells to the primary lesion core...”. This claim is very much an overstatement and should be appropriately qualified. At best microglial play a role in the multicellular interactions that serve sequester immune cells in the lesion core. The study here presents no evidence that microglial do so on their own, or that this is their primary role after SCI, and the current sentence implies both of these things. The wording also seems very reminiscent of what has been reported by others regarding the astroglia scar. The authors should modify this ending sentence to indicate the multicellular nature of this regulation and cite appropriate references in this regard.

The text has been modified according to recommendations made by the reviewer (page 23, lines 546-549): “In light of the current data, we conclude that activated, proliferating microglia play a key role in the formation of the scar that develops after SCI, and that these multicellular interactions serve to sequester blood-derived immune cells in the lesion core, thus protecting non-injured neurons and oligodendrocytes from inflammation-mediated tissue damage.”

Reviewer 3:

The manuscript by Bellver-Landete and colleagues describes the microglial cell reaction after a spinal cord injury. The authors study the still poorly understood reaction of microglia and their subsequent activation after injury. Using CX3CR1-CreERT2/R26-tdT mice, the authors take advantage of the different proliferation rates of CNS microglia and peripheral myeloid cells to selectively label microglial cells in the spinal cord, but not monocyte-derived macrophages. Microglia are highly active, proliferate in the first weeks and generate a ‘microglial scar’. Pharmacological depletion of microglia did not only lead to disrupted scar formation, but also to impaired locomotor recovery. In general, this study is concisely designed and well written. The results are novel and important. With these observations, the authors provide further insights into the mechanisms of spinal cord injury and its repair. Furthermore, they offer novel routes to develop therapeutic approaches.

- 1) The immunohistochemical images are hard to appreciate. The authors should display the individual color channels to grey scale and present color views of merged channels (for example Fig. 1a-f

(cyan/red) or Suppl. Fig. 5a-c (blue, green, red, cyan)). In addition, the x-axis indicating the distance from the injury center should be labeled by rounded numbers, e.g. 50, 100, 150 μm .

The individual color channels are now presented for all figures (**Figs. 1, 5 & 6**), with some of them available as supplementary figures (**Suppl. Figs. 2, 3, 4, 5, 7, 8, & 10**). As suggested by Reviewer 3, the distances from the lesion epicenter have been rounded up to the nearest hundredth.

2) In Fig. 2 J, at dpi 7 LysM eGFP⁺ cells were largely reduced in PLX5622-treated mice. The authors “interpret this to mean that the delayed myeloid cell recruitment in the injured spinal cord of PLX5622-treated mice was caused by the absence of microglia rather than a direct effect on peripheral leukocytes” (Line 198-200). However, the number of tdT⁺ microglia in PLX73086-treated mice was also reduced by half at dpi 7 (Fig. 2f), while the LysM eGFP⁺ cells were not altered. These results could be explained as well by the two drugs (PLX5622 and 73086) having different effects on the infiltration of myeloid cells into the lesion site after SCI. Please address.

PLX5622 and PLX73086 are selective inhibitors of CSF1R tyrosine kinase activity with comparable potency against CSF1R at the concentrations used in our study (Dr. Andrey Rymar, Plexxikon, personal communication). In contrast to PLX3397, these two drugs do not inhibit c-Kit (Kim et al., *Clin Cancer Res*, 2014), a receptor essential to the maintenance and survival of hematopoietic stem cells. Since CSF1R was previously shown to be dispensable for the survival of cells of the monocyte/macrophage lineage and their egress from the bone marrow into the blood circulation in response to sterile inflammation (Hibbs et al., *J Immunol*, 2007), which is in agreement with our data presented in **Suppl. Fig. 6**, we argue that the different effects of PLX5622 and PLX73086 on the infiltration of myeloid cells into the site of SCI have to be explained by the greater efficiency of PLX5622 in depleting microglia. As we previously showed, microglia are one of the primary cellular sources of monocyte chemoattractants at the site of SCI, with the peak of cytokine and chemokine production occurring within the first 24 hours (Pineau et al., *J Comp Neurol*, 2007). In the present study, treatment of C57BL/6 mice with PLX5622 resulted in depletion of ~98% of spinal cord microglia at day 1 post-SCI, while PLX73086 failed to deplete microglia at that time (**Fig. 2a-d**). Taking all these results together, we feel confident speculating that the reduction in the number of LysM-eGFP⁺ cells at the lesion epicenter at day 7 in the PLX5622 group is the consequence of the compromised early response of microglia.

3) The main conclusion of this study is, that in the first week after SCI, microglia are activated and proliferate drastically to form the so-called ‘microglia scar’ which is pivotal for the functional recovery

of the spinal cord. However, this conclusion is mainly based on the cell ablation results using the CSF-1R inhibitor PLX5622. However, in addition to microglia, myeloid cells also express CSF-1R, therefore the functional recovery effect could also be the total outcome of combining the inhibitory effect on microglia and myeloid cells, which is also suggested by the reduction of LysM EGFP⁺ cells in the first week (also see point 1). The authors claim that the “reduction in myeloid cell infiltration was however transient and overcome by day 14”, that’s exactly the important time window. The authors should provide more precise evidence to distinguish the functional outcome of inhibiting microglia and myeloid cells, respectively.

As discussed above, we claim that inhibition of CSFR1 activity by PLX5622 had a minimal direct effect on the monocyte/macrophage population, as supported by **Suppl. Fig. 6**. However, as rightly pointed out by Reviewer 3 and shown in **Fig. 2k**, microglial depletion significantly reduced myeloid cell recruitment in the injured spinal cord during the important time window (i.e. the first week post-SCI). Although evidence indicates that the reduced infiltration of myeloid cells was a direct consequence of the absence of microglia (see response to Comment #2), the fact of the matter remains that myeloid cells may have also contributed to the functional recovery effect. This is now clearly stated in the revised manuscript (page 21, lines 492-497): “Although the early infiltration of blood-derived myeloid cells at sites of SCI was previously associated with neurotoxicity^{45, 46, 47}, we cannot rule out the possibility that these cells may have contributed to the functional recovery effect seen in PLX5622-treated mice. If it were to be the case, we argue that it would be under the positive influence of microglia as evidence here indicates that the reduction in myeloid cell infiltration was a direct cause of the absence of microglia.”

45. Blight AR. Effects of silica on the outcome from experimental spinal cord injury: implication of macrophages in secondary tissue damage. *Neuroscience* **60**, 263-273. (1994).
46. Popovich PG, Guan Z, Wei P, Huitinga I, van Rooijen N, Stokes BT. Depletion of hematogenous macrophages promotes partial hindlimb recovery and neuroanatomical repair after experimental spinal cord injury. *Exp Neurol* **158**, 351-365. (1999).
47. Kigerl KA, Gensel JC, Ankeny DP, Alexander JK, Donnelly DJ, Popovich PG. Identification of two distinct macrophage subsets with divergent effects causing either neurotoxicity or regeneration in the injured mouse spinal cord. *J Neurosci* **29**, 13435-13444 (2009).

4) Fig. 3g-i: why are respective experiments with PLX73086 missing?

The reason why this group was not included is because Plexikon was unable at the time to provide us with the drug in suspension.

5) Why are the BMS subscores in Fig. 3l lower than in other experimental conditions?

Regarding our behavioral analyses, we like to point out that they are always performed blind with respect to the identity of the animals by an Animal Research Technician who has >15 years of experience using the Infinite Horizon impactor device and >10 years of experience with the Basso Mouse Scale (BMS). We also emphasize that we thrive on generating experimental results that are highly reproducible. Based on the maximum standard deviation observed in our previously published work reporting recovery of locomotor function over time after SCI, we have calculated that 10 mice per group will give us >98% probability of detecting a significant change if alpha is set at 0.05 and standard deviations are 20% of average. Accordingly, a *n* of at least 10 mice per group was always used for all behavioral analyses performed in this study. Nevertheless, *in vivo* studies are not immune to some form of heterogeneity, which is why we have included a control group in every single behavioral experiment.

6) The depletion of microglia (with PLX5622) caused increased oligodendroglial and neuronal cell death after injury. Where are the controls in which healthy mice were treated with PLX5622 and PLX73086?

We apologize for this omission. These controls have now been added to the **revised Fig. 8** (see graphs in **l** and **p**).

7) The authors regard day 7 after injury as the most important time point. In the M-CSF hydrogel experiments, however, the BMS scores of the PBS control group (Fig. 6e and f) are much lower at day 7 than the scores in other control experiments (Fig. 3). Why?

Please see response to Comment #5 (above).

8) Please add the precise time points for the drug treatment (Fig. 2 d-g and also in Suppl. Figures).

As requested by Reviewer 2 (see Comment #6) and Reviewer 3, the treatment paradigm (route of administration, duration, etc.) is now explicitly stated in the *Results* section as well as in the legend of Figure 2. The text was modified as follows to reflect these modifications: 1) Page 8, lines 171-172: “Mice were fed chow containing either PLX5622, PLX73086 or vehicle (without gavage) starting 3 weeks before SCI and until time of sacrifice.”, and 2) Page 45, lines 1068-1069: “For all injured mice, treatment was initiated 3 weeks before SCI and continued until sacrifice.”

9) In line 238 the authors claim that the number of P2ry12⁺ microglia increase after treatment (>54 %). Where is that dataset? In Suppl.-Fig. 5, no P2ry12⁺ staining is mentioned.

The text has been revised to indicate that these data are not shown (page 11, lines 235-238): “A similar trend was also seen in C57BL/6 mice, in which the total number of P2ry12⁺ microglia, after a repopulation period of 7 days, exceeded that of untreated mice by 54% (116.5 ± 5.4 P2ry12⁺ cells/mm² compared to 75.7 ± 2.1 P2ry12⁺ cells/mm²; data not shown).”

10) Suppl. Fig. 5 is supposed to show the microglia morphology after repopulation (“Repopulated TdT⁺ cells seen after 7 days of drug withdrawal had a ramified morphology and expressed CD11b and Iba1 (Suppl. Fig. 5a-c, e)”). However, this is very hard to recognize at the images a to c. Please provide magnified views of exemplary cells. Olig2 is not the perfect marker for the oligodendrocyte lineage in disease models. It might also be upregulated in activated astrocytes. Please use other markers of the oligodendrocyte lineage such as PDGFRalpha (OPC) or CC1 (mature oligodendrocytes).

As requested, the **revised Suppl. Fig. 7** (Suppl. Fig. 5 in the original submission) now includes representative examples of magnified views of repopulated microglia. Additionally, we now provide quantification of Olig2⁺ CC1⁺ mature oligodendrocytes in the **modified Fig. 8**.

11) In Fig. 6 (line 388), the authors claim that the M-CSF delivering hydrogels strikingly reduce the lesioned area. And yes, the bar in 6d is smaller. But the data do not reach statistical significance. Please rephrase or provide more data.

The reviewer is right. Please see response to Comment #12 of Reviewer 2, who also raised the same concern about conclusions that were based on data that did not reach statistical significance. This issue has been solved by increasing the number of mice in each group and by clearly indicating whether the data are significant or not. Accordingly, the following statement has been clarified as follows (page 18, lines 418-421): “Despite the fact that treating *Cx3cr1*^{creER::R26-TdT} mice with the M-CSF-based hydrogel resulted in a non-significant trend towards a higher number of TdT⁺ microglia (Fig. 9c), it was sufficient to reduce the lesion area rostral to the injury epicenter at 7 dpi (Fig. 9d).”

12) Please delete in line 354 “counted significantly fewer neurons” (written twice).

Done.

13) Methods part: Please clarify which mice were used: C57Bl/6N or J?

C57BL/6N mice were used. The information has been corrected in the *Methods* section (page 24, line 551).

In summary, the paper shows very interesting new data addressing a unique function of microglia in the spinal cord.

Once again, we appreciate the comments of the reviewers and hope our responses and revisions satisfactorily address their points. Please do not hesitate to contact us if additional information is required.

Sincerely,

Steve Lacroix, Ph.D.

Director of the Neurosciences Axis, CHU de Québec–Université Laval Research

Center Full Professor, Department of Molecular Medicine

Faculty of Medicine, Laval University

2705, Boul. Laurier, local T2-50

Québec, Québec

Canada, G1V 4G2

REVIEWERS' COMMENTS:

Reviewer #1 (Remarks to the Author):

I`m happy with the revision and have no further comments left.

Reviewer #2 (Remarks to the Author):

The authors have done commendable job in revising their paper. They have made substantial substantial edits and additions to their manuscript. They have appropriately dealt with all of my comments and suggestions. I have no more concerns. I think that this is now a strong paper that will be of wide interest and importance in the spinal cord injury field and beyond that with respect to areas of other forms of injury and disease in the central nervous system.

Reviewer #3 (Remarks to the Author):

The authors addressed my comments on their manscript. In addition, they provide more and novel data.

This is an important study highlighting the direct impact of microglia in spinal cord injury at the interface of invading leukocytes and scar-forming astrocytes. The reciprocal regulation of microglia depletion, stimulation by M-CSF1 or establishing IGF-1 to affect astrocytes are findings that will stimulate further work.

Steve Lacroix, Ph.D.

Full Professor

December 20th, 2018

RE: Manuscript # NCOMMS-18-24784B

Dear referees,

We are pleased that our manuscript entitled “**Microglia are an essential component of the neuroprotective scar that forms after spinal cord injury**” has been found acceptable for publication in *Nature Communications*.

The three referees had no further concerns and all recommended acceptance of the paper. We thank them all for their constructive comments.

Reviewer 1:

1) “I’m happy with the revision and have no further comments left.”

The modifications suggested by Reviewer 1 have significantly improved the final version of our study by increasing its scientific quality and relevance, and we are very thankful for that.

Reviewer 2:

1) “The authors have done commendable job in revising their paper. They have made substantial edits and additions to their manuscript. They have appropriately dealt with all of my comments and suggestions. I have no more concerns. I think that this is now a strong paper that will be of wide interest and importance in the spinal cord injury field and beyond that with respect to areas of other forms of injury and disease in the central nervous system.”

We agree and thank Reviewer 2 for his constructive comments that helped us to improve our manuscript.

Reviewer 3:

1) “The authors addressed my comments on their manuscript. In addition, they provide more and novel data. This is an important study highlighting the direct impact of microglia in spinal cord injury at the interface of invading leukocytes and scar-forming astrocytes. The reciprocal regulation of microglia depletion, stimulation by M-CSF1 or establishing IGF-1 to affect astrocytes are findings that will stimulate further work.”

The positive feedback received from Reviewer 3 is truly appreciated.

Once again, we appreciate the comments of the referees and are delighted that our previous responses and revisions satisfactorily addressed their points. Please do not hesitate to contact us if additional information is required.

Sincerely,

Steve Lacroix, Ph.D.

Director of the Neurosciences Axis, CHU de Québec–Université Laval Research Center

Full Professor, Department of Molecular Medicine

Faculty of Medicine, Laval University

2705, Boul. Laurier, local T2-50

Québec, Québec

Canada, G1V 4G2